

# Atmospheric CO₂ inversions at the mesoscale using data driven prior uncertainties. Part1: Methodology and system evaluation

Panagiotis Kountouris[1], Christoph Gerbig[1], Christian Rödenbeck[1], Ute Karstens[1,*], Thomas F. Koch[2], Martin Heimann[1]

[1]Max Planck Institute for Biogeochemistry, Jena, Germany

[2]Meteorological Observatory Hohenpeissenberg, Deutscher Wetterdienst, Germany

[*]Now at ICOS Carbon Portal, Lund University, Lund, Sweden

*Correspondence to*: P. Kountouris (pkount@bgc-jena.mpg.de)



# 1 Abstract

Atmospheric inversions are widely used in the optimization of surface carbon fluxes at regional scale using information from atmospheric $CO_2$ dry mole fractions. In many studies the prior flux uncertainty applied to the inversion schemes does not reflect directly the true flux uncertainties but it is used in such a way to regularize the inverse problem. Here, we aim to implement an inversion scheme using the Jena inversion system and applying a prior flux error structure derived from a model – data residual analysis using high spatial and temporal resolution over a full year period in the European domain. We analyzed the performance of the inversion system with a synthetic experiment, where the flux constraint is derived following the same residual analysis but applied to the model-model mismatch. The synthetic study showed a quite good agreement between posterior and "true" fluxes at European/Country and annual/monthly scales. Posterior monthly and country aggregated fluxes improved their correlation coefficient with the "known truth" by 7% compared to the prior estimates when compared to the reference, with a mean correlation of 0.92. Respectively, the ratio of the standard deviation between posterior/reference and prior/reference was also reduced by 33% with a mean value of 1.15. We identified temporal and spatial scales where the inversion system maximizes the derived information; monthly temporal scales at around 200 km spatial resolution seem to maximize the information gain.





# 1 Introduction

The continuous rise of the abundance of greenhouse gases in the atmosphere, especially due to fossil fuel combustion, alerted the scientific community to systematically monitor these emissions. The challenge is not limited only to revealing the spatial distribution of $CO_2$ sources and sinks on continental scales, but also to accurately quantifying $CO_2$ emissions and their uncertainties at country scales. In situ atmospheric measurements of the atmospheric $CO_2$ variability combined with inverse atmospheric models are used as an independent method to provide "top down" flux estimates for comparison with estimates from "bottom up" methods. The latter use local observations (e.g. eddy covariance), and combine these with ancillary data, e.g. soil maps, satellite data, and terrestrial ecosystem models in order to spatially scale up local flux estimates to larger regions (Jung et al., 2009). Both approaches act complementary, for optimal comprehension of carbon sources and sinks in a "multiple constraint" (Schulze et al., 2010) approach and emission inventories assessment. As these inventories are used to deduce national emission estimates, in compliance with the Kyoto protocol requirements, accuracy is essential.

An atmospheric inverse modeling system provides the link from atmospheric concentrations to surface fluxes. However, the limited number of observations available for solving the system for quite a number of unknowns (spatially and temporally resolved fluxes) makes the inverse problem strongly under-determined. To solve the inverse problem the system incorporates Bayes' theorem and uses a-priori knowledge, provided by e.g. biosphere models and emission inventories accompanied by corresponding uncertainty estimates. Then, the system optimizes the a-priori fluxes by minimizing the difference between model predictions and observed concentrations. For the current study only the biospheric fluxes were optimized, and emissions from fossil fuel combustion are assumed to be known much better, as it is the case in almost all published regional inversion studies. Inversion systems have been extensively used to derive spatiotemporal flux patterns at global (e.g. Enting et al., 1995; Kaminski et al., 1999a; Gurney et al., 2003; Mueller et al., 2008), and regional scale (e.g. Gerbig et al., 2003a; Peylin et al., 2005; Lauvaux et al., 2012; Broquet et al., 2013).



The challenge in regional inversions is to reconstruct at high resolution the spatiotemporal flux
patterns, usually of the net ecosystem exchange (NEE). For that purpose currently deployed
global or regional inverse modeling schemes use different state spaces (i.e. the set of variables to
be optimized through the inversion process). Peters et al. (2007) split the domain of interest into
regions according to ecosystem type. Subsequently fluxes are optimized by using linear
multiplication factors to scale NEE for each week and each region. The pitfall of this system is
that a zero prior flux has no chance to be optimized and remains zero. Zupanski et al. (2007)
divided the NEE into two components, i.e. the gross photosynthetic production (GPP) and
ecosystem respiration (R). Then multiplicative factors for the gross fluxes were derived on the
grid scale, under the assumption of being constant in time. A step further made by Lokupitiya et
al. (2008) used the same approach but with an 8-week time window allowing for temporal
variations for the multiplicative factors. A different approach introducing the carbon cycle data
assimilation system (CCDAS) was implemented by Rayner et al. (2005) and Kaminski et al.
(2012) by constraining global parameters within a biosphere model able to control surface-
atmosphere exchange fluxes, against observed atmospheric $CO_2$ mole fractions, instead of the
fluxes themselves. Lauvaux et al. (2012) used a Bayesian approach based on matrix inversion,
separately optimizing day and night time fluxes at a weekly time scale for a limited simulation
period and domain. An attempt to assess which of these approaches better reproduces NEE was
made by Tolk et al. (2011). This study investigated the impact of different inversion approaches
via a synthetic experiment utilizing an ensemble Kalman filter technique and the same transport
model for all cases. They found that inversions which separately optimize gross fluxes within a
pixel inversion concept perform better on reconstructing the NEE, although they fail to obtain
the gross fluxes. Taking into consideration these findings we also choose the pixel based
inversions but optimizing the net biogenic fluxes as we are mainly interested in the total carbon
flux budget.
Introducing proper prior flux uncertainties is crucial for meaningful posterior estimates, as these
uncertainties weight the prior knowledge between different locations and times, as well as with
respect to the data constraint. The uncertainties have the form of a covariance matrix and can be
categorized in uncertainties of the prior fluxes, and uncertainties of the observational constraint,
which includes measurement and transport model uncertainties. While the observational
constraint may be more easily defined with the main diagonal of the covariance matrix



representing the uncertainty of the observations and the model at a specific time and location, our
knowledge for the prior uncertainty is limited. Early inversions assumed fully uncorrelated flux
uncertainties (Kaminski et al., 1999b), while spatial and temporal correlations were used later by
Rödenbeck et al. (2003), who investigated the autocorrelation of monthly $CO_2$ fluxes calculated
by a set of terrestrial and ocean models. In Rödenbeck (2005), spatial correlations for land fluxes
were assigned to a state space of 4° latitude x 5° longitude resolution. Slightly different
correlation length scales were considered for the meridional and zonal direction, assuming that
the climate zone of the later varies less than of the former. Flux correlations on land were
determined by assuming an exponential pulse response function with a length of 1275 km. This
leads to correlations with approximately twice the correlation length. Typically the spatial
correlations are considered more as a tool to regularize the inverse problem, rather than an
uncertainty feature. Schuh et al. (2010) obtained correlation lengths from Rödenbeck et al.
(2003) but with a much higher state space resolution of 200 km.  Lauvaux et al. (2008) neglected
the spatial correlations to enlarge the impact of the data. Carouge et al. (2010a) inferred spatial
and temporal correlation lengths based on the agreement between posterior and "true" fluxes in
the framework of a synthetic experiment, where the "truth" is known. A different approach was
used in Peters et al. (2007) study where they interpret the length scale from a climatological and
ecological perspective, and use it to spread information within regions, which the network is
incapable to constrain. Ad-hoc solutions have also been used, assuming that daily fluxes have
smaller correlation lengths than monthly fluxes which are used by other studies (Peylin et al.
2005). More specifically Peylin et al. (2005) assumed 500 km for daily temporal resolution
compared to the much larger correlation lengths used by Rödenbeck for monthly flux resolution.
Michalak et al. (2004) implemented a geostatistical approach to describe the prior error structure.
Specifically the prior error covariance describes at which degree deviations of the surface fluxes
from their mean behavior at two different locations or times are expected to be correlated as a
function of the distance in space or in time. They simultaneously estimate posterior fluxes as
well as parameters controlling the model-data mismatch uncertainty and the prior flux
uncertainty, including spatial and temporal correlation lengths. Although this approach may be
considered as an objective way to infer spatial and temporal correlation lengths, it forces the
error covariance to be statistically consistent with the atmospheric data from the few regions
where station-to-station distances are small enough to be comparable to the correlation length





scales. Eddy Covariance stations (EC) can provide a more direct method to infer spatial and
temporal flux correlations. Chevallier et al. (2006) and Chevallier et al. (2012) introduced
autocorrelation analysis of the residual between fluxes simulated by biosphere models or
measured by EC to infer spatial and temporal error correlations. The derived error statistics were
implemented in a regional $CO_2$ inversion by Broquet et al. (2013).
Daily NEE flux residuals from model - data comparisons showed temporal correlations up to 30
days but very short spatial correlations up to 40 km (Kountouris et al. 2015). In such a case the a-
priori integrated uncertainty over time and space, e.g. annually and EU wide domain integrated,
according to the error propagation will be exceptionally small. For example a variance of 1.82
$\mu mole.m^{-2}.s^{-1}$ (from model – data differences) combined with the abovementioned correlation
scales yields an uncertainty of 0.12 GtC $y^{-1}$ for the total flux over Europe. This value is
significantly smaller than the assumed uncertainty which is typically used by the inversion
systems. For comparison we refer to studies from Rivier et al. (2010) and Peylin et al. (2005) (for
a slightly larger domain than ours) where an a priori uncertainty of approximately 1.4 GtC $y^{-1}$
and 1 GtC $y^{-1}$ respectively was used. Further, Peylin et al. (2013) found that the variance of the
posterior NEE fluxes for the European domain among 11 global inversions is also 3 to 4 times
larger (0.45 GtC $y^{-1}$). Although is not yet entirely clear what would be the "correct" value for the
prior uncertainty, it seems that in our study it should be increased not only to give enough
flexibility to the system to adjust but also to be at least comparable with other posterior
uncertainty estimates. A typical method is to inflate the spatiotemporal component by scaling
accordingly the prior error covariance. In a study by Lauvaux et al. (2012) two correlation
lengths were used at 300 and 50 km, and for the shorter scale the uncertainty was inflated by
increasing the RMS of the prior error covariance. The model - data analysis (Kountouris et al.
2015) does neither justify the use of large correlation scales nor largely inflated variances which
exceed the model-data flux mismatches, however it is consistent with an additional overall bias
error which can not be captured from the estimated spatiotemporal error structure. Hence an
appropriate approach would be to introduce two adjustable terms into the inversion system. One
term to reflect the data-derived error structure without error inflation (prior error covariance
matrix which describes the spatiotemporal component) and one term to represent a bias
component. To the best of our knowledge such an approach has not yet been used in inversion
systems.





This study primarily aims to use the information extracted from the model-EC data residuals
(spatiotemporal error structure) to define a data-driven error covariance rather than simply
assuming one, adopting a conservative one or an expert knowledge solution. For that, we
implement our previous methodology and findings regarding the prior uncertainty to atmospheric
inversions following Kountouris et al. (2015). As explained above, we implement two
uncertainty terms; the first one to reflect the true spatiotemporal error structure and the second
term referred to a bias term. We use the Jena inversion system (Rödenbeck, 2005; Rödenbeck et
al., 2009) for the regional scale consisting of a fully coupled system as described in Trusilova et
al. (2010), between the global three-dimensional atmospheric tracer transport model TM3
(Heimann and Körner, 2003) and the regional stochastic Lagrangian transport model STILT (Lin
et al., 2003). This scheme allows retrieving surface fluxes at much finer resolution ($0.25^{o}$)
compared to global models. The first part of this study details the methodology of the prior error
implementation, and evaluates the system's performance through a synthetic data experiment.
The system evaluation is an extension of Trusilova et al. (2010) where the evaluation was limited
to the observation space only. We extend that to the flux space by comparing flux retrievals at
various spatial and temporal scales against synthetic "true" fluxes. Station locations and
observation times (including gaps) were created as in the real observation time series presented
in the second part of this study (Kountouris et al., 2016). That way we can use the synthetic
experiment to evaluate to what extent we can trust the results, if a real-data inversion is
performed. In the second part of this study (Kountouris et al., 2016) the regional inversion
system is applied to real observations of atmospheric $CO_2$ mole fractions from a network of 16
stations.
This paper is structured as follows. In Section 2 we present the inversion scheme and introduce
the settings of the atmospheric inversions. In Section 3 we present the results from a synthetic
inversion experiment aimed to assess the prior error setup, considering it as a step towards
atmospheric inversions using real atmospheric data with an objective, state of the art prior error
formulation. Discussion and conclusions are following in Section 4.



## 2 Methods

### 2.1 Inversion scheme

The Jena Inversion System (Rödenbeck 2005; Rödenbeck et al., 2009) was used for the current study. The scheme is based on the Bayesian inference and uses two transport models, the TM3 model (Heimann and Körner, 2003) for global, and the STILT model (Lin et al., 2003) for regional simulations. The advantage of the system is that it combines a global transport model with a regional one without the need of a direct coupling along the boundaries. The global is used to calculate fluxes from the far field (outside of the regional domain of interest), and subsequently this information can be used to provide lateral boundary information for the regional model. Primary input of the system is the observed mixing ratios $c_{meas}$. This vector contains all measured mixing ratios at different times and locations. The modeled mixing ratios $c_{mod}$ given from a temporally and spatially varying discretized flux field $f$ are computed from an atmospheric transport model and can be formally expressed as

$$c_{mod} = Af + c_{ini} \tag{1}$$

where $c_{ini}$ is the initial concentration and $A$ the transport matrix which maps the flux space to the observation space. For the regional domain the transport matrix $A$ has been pre-computed by the STILT transport model. The system calculates the modeled concentrations when and where a measurement exists in the $c_{meas}$ vector.

In the following, we briefly describe the inverse modeling approach. For more details the reader is referred to Rödenbeck (2005).

In grid-based atmospheric inversions the number of unknowns (spatially and temporally resolved fluxes) is larger than the number of measurements (hourly dry mole fractions at different sites), making the inverse problem ill-posed. In the Bayesian concept this can be remedied by adding a-priori information. This information can be written as

$$f = f_{fix} + F \cdot p \tag{2}$$





where $f_{fix}$ is the a-priori expectation value of the flux, matrix $F$ contains all the a-priori
information about flux uncertainties and correlations (implicitly defining the covariance matrix)
and $p$ is a vector representing the adjustable parameters. The parameters $p$ are uncorrelated with
zero mean and unit variance. This flux model represents just a different way to define the a-priori
probability distribution of the fluxes, than the traditional way where the a-priori error covariance
matrix is explicitly specified. The cost function describing the observational constrain is
expressed as
$$J_c = \frac{1}{2}(c_{meas} - c_{mod})^T \cdot Q_c^{-1} \cdot (c_{meas} - c_{mod})$$ (3)
where $Q_c$ is the observation error covariance matrix. This diagonal matrix weights the mixing
ratio values considering measurement uncertainty, location-dependent model uncertainty and a
data density weighting. The latter ensures that the higher amount of data from continuous
measurements compared to the data from flask measurements would not lead to a considerably
stronger impact of these corresponding sites (Rödenbeck, 2005). This can also be formally
interpreted as a temporal correlation scale which ensures that the model-data-mismatch error is
not independent within a week, corresponding roughly to time scales of synoptic weather
patterns.
The inversion system seeks to minimize the following cost function that combines the
observational (Eq. 3) and the prior flux constrain
$$J = J_c + \frac{1}{2} \cdot p^T \cdot p$$ (4)
The minimization of the cost function is done iteratively with respect to the parameters $p$ by
using a Conjugate Gradient algorithm with re-orthogonalization (Rödenbeck 2005).



## 2.2 Characteristics of the inversion set up

### 2.2.1 A-priori information and uncertainties

The a-priori $CO_2$ flux fields were derived from the Vegetation Photosynthesis and Respiration Model, VPRM (Mahadevan et al., 2008). VPRM uses ECMWF operational meteorological data for radiation (downward shortwave radiative flux) and temperatures (T2m), the SYNMAP landcover classification (Jung et al., 2006), and EVI (enhanced vegetation index) and LSWI (land surface water index) derived from MODIS (Moderate Resolution Imaging Spectroradiometer). Model parameters were re-optimized for Europe using eddy covariance measurements made during 2007 from 47 sites (a full site list is given in Kountouris et al. (2015); we excluded some sites due to insufficient temporal data coverage or lack of representativeness). To mediate the impact of data gaps, a data density weighting was introduced that takes into account the coverage of different times of the day (using 3-hour bins) in the different seasons. Optimized parameters are shown in Table 1. The net ecosystem exchange at hourly scale and at $0.25°$ x $0.25°$ spatial resolution for 2007 was simulated with the optimized parameters for the European domain shown in Fig. 1. The domain-wide aggregated biospheric carbon budget for 2007 derived that way from VPRM was found to be -0.96 GtC $y^{-1}$ (i.e. uptake by the biosphere). Note that without the density weighting an even stronger flux of -1.35 GtC $y^{-1}$ was derived, indicating the importance of proper treatment of data gaps by either gap-filling or by the inclusion of weights.

Additionally, biogenic $CO_2$ fluxes were simulated with the BIOME-BGC model, specifically its global implementation as GBIOME-BGCv1 (Trusilova and Churkina 2008) at the same $0.25°$ x $0.25°$ spatial and hourly temporal resolution. The purpose of the second flux field is to provide a perfectly known flux distribution as "true" fluxes that can be used to generate synthetic observations. The a-priori flux in a real-data inversion would have three components including fossil fuel and ocean fluxes

$$f_{pr} = f_{pr,nee} + f_{pr,ff} + f_{pr,oc} \tag{5}$$



We note that for the synthetic case the last two terms are set to zero. Similarly the deviation term
(the data-derived correction to the a-priori fluxes) of the flux model consists of the terms
referring to NEE, fossil fuel, and ocean fluxes but equivalently the last two terms are set to zero
for the synthetic inversion.

$$F \, \delta s = (F_{nee} \, , F_{oc} \, , F_{ff} \,) \begin{pmatrix} \delta s_{nee} \\ \delta s_{oc} \\ \delta s_{ff} \end{pmatrix} \qquad (6)$$

Note that the a-priori error covariance matrix does not explicitly appear in the inversion, but is
included though the second term in Eq. 8 (see section 2.2.2).
According to this formulation the columns of $G_{tcor}$ and $G_{xycor}$ contain the spatiotemporal extents
of the individual NEE pulses (range of values between 0 and 1) and the diagonal matrix $f_{sh}(x,y,t)$
contains the pixel-wise a priori uncertainties. These uncertainties were chosen to be flat
(constant) in space and time. For more detailed information the reader is referred to Rödenbeck
et al. (2005).
The total prior uncertainty was chosen according to the mismatch between VPRM and BIOME-
BGCv1, calculated as the annual and domain wide integrated flux mismatch. Prior fluxes and the
fluxes representing the synthetic truth are strongly different (-0.96 GtC y$^{-1}$ and -0.31 GtC y$^{-1}$ for
VPRM and GBIOME-BGCv1, respectively). The error structure used for the synthetic study is
estimated according to the method applied in Kountouris et al. (2015). Time-series of daily
fluxes were extracted for both biosphere models at grid cell locations where an EC station exists.
Then spatial and temporal autocorrelation analysis was performed on the daily model-model flux
residuals, yielding a spatial correlation length scale of 566 km and a temporal correlation scale of
30 days.
The eddy covariance station locations used for this analysis were exactly the same as in
Kountouris et al. (2015) ensuring similarity in the derivation of the error structure for the
synthetic data inversions. However of note is that for the synthetic data inversions, prior fluxes
from VPRM model were not optimized against GBIOME-BGCv1 "true" fluxes.



The implicitly defined prior error covariance matrix contains diagonal elements of $(1.45\ \mu mol\ m^{-2}\ s^{-1})^2$, which reflect the variance from model-model flux mismatches at the 50 km spatial resolution of the state space. Exponentially decaying spatial correlations were implemented with a correlation scale of 766 km at the zonal and 411 km at the meridional direction, roughly corresponding to the 566 km correlation scale yielded from the model-model residual autocorrelation analysis and preserving the same zonal/meridional ratio as in the global inversion. Temporal autocorrelation was set to 31 days, which is consistent with the Kountouris et al. (2015) analysis. These scales result in an uncertainty for the spatiotemporal component ($E_{st}$) domain-wide and annually integrated of 0.44 GtC $y^{-1}$. We chose two different approaches to increase the prior uncertainty at domain-wide and annually integrated scale such that it matches the mismatch of 0.65 GtC $y^{-1}$ between the two biosphere models. First we inflate the error by scaling the error covariance matrix, this case is referred to as base case B1 hereafter. The second approach, referred to as scenario S1, could be considered as a more formal way: we introduce an additional degree of freedom to the inversion system by allowing for a bias term. This term is spatially distributed according to the annually averaged VPRM respiration component, and is kept constant in time. The error $E_{BT}$ of the bias component was adjusted such that the total prior error $E_{tot}$ for annually and domain-wide integrated fluxes matches the targeted total uncertainty:

$$E_{tot}^2 = E_{ST}^2 + E_{BT}^2 \qquad (7)$$

This resulted in an overall uncertainty $E_{tot}$ of 0.65 GtC $y^{-1}$, which is identical to the mismatch between the two biosphere models.

### 2.2.2 State space

The inversion system optimizes additive corrections to three-hourly fluxes in a sense that the posterior flux estimate can be given by the sum of a fixed a priori term (first term of the right hand side in Eq. 8) and an adjustable term (second term in Eq. 8). The latter has a-priori a zero mean and unit variance. The biogenic fluxes can be defined as follows:





$$f(x,y,t) = f_{fix}(x,y,t) + f_{sh}(x,y,t) \cdot \sum_{m_t}^{N_t} \sum_{m_s}^{N_s} G_{tcor,m_t}(t) \cdot G_{xycor,m_s}(x,y) \cdot p_{inv,m_t,m_s} \qquad (8)$$
where $f_{sh}$ is a shape function which defines the adjustable term. The spatial and temporal
correlation structures of the uncertainty are described by the pulse response functions $G_{xycor}$ and
$G_{tcor}$ respectively. The term $p_{inv}$ contains the adjustable parameters which they a-priori have, a
Gaussian distribution with zero mean and unit variance.
For the S1 case the posterior flux estimates can be derived by adding the optimized bias flux
field to Eq. 8
$$f(x,y,t) = f_{fix}(x,y,t) + f_{sh}(x,y,t) \cdot \sum_{m_t}^{N_t} \sum_{m_s}^{N_s} G_{tcor,m_t}(t) \cdot G_{xycor,m_s}(x,y) \cdot p_{inv,m_t,m_s} + f_{sh}^{BT}(x,y) \cdot \sum_{m_t}^{N_t} G_{tcor,m_t}(t) \cdot p_{BT}$$
$\qquad (9)$
The bias term $f^{BT}$ follows a flux shape (here we used annually averaged respiration, with no
temporal variation).
**2.2.3 Observation vector and uncertainties**
The observation vector $c_{meas}$ contains mixing ratio observations at all site locations and sampling
times. A common procedure to derive synthetic observations is to create a "true" flux field by
adding some error realizations to the a-priori fluxes (Schuh et al., 2009; Broquet et al., 2011) or
to perturb the resulting synthetic observations (Wu et al., 2011). For the current study instead we
use a different biosphere model, the GBIOME-BGCv1 model, to derive biogenic $CO_2$ fluxes at
hourly scale. Then a forward transport model run was performed to create synthetic mixing ratios
at hourly resolution for each station location. This choice of using two different biosphere
models for deriving the a-priori and the "true" fluxes is expected to increase the realism of the
synthetic data study, given the fact that the real spatiotemporal flux distribution is highly
unknown (though the model-to-model difference may not accurately reflect the model errors
either). For the synthetic study, observations were created for the same station locations and





observation times as in the real observation time series which are used in the second part of this
study (Kountouris et al., (2016)). An overview of the atmospheric stations is given in table 2.
The data coverage per station is shown in Figure 2. Only daytime observations were considered
(11:00 – 16:00 local time) since the transport model is expected to perform worse during night
when a stable boundary layer forms. An exception is made for mountain stations that measure
the free troposphere, where only nighttime observations (23:00 – 04:00 local time) were
considered, as this time can be better represented by the transport model. In total 20273 hourly
observations from the year 2007 were used.
The model-data mismatch uncertainty associated with each measurement is expressed as a
diagonal covariance matrix, and contains measurement errors and errors from different
components describing the modeling framework (i.e. model errors due to imperfect transport,
aggregation errors, etc.) (Gerbig et al., 2003b). For the current study, all sites are classified
according to their characteristics (e.g. tall tower, mountain sites etc.), and uncertainties were
defined depending on the site class (Figure 2, legend on the right). The uncertainties are
considered as representative for current inverse modeling systems. Although the measurement
error covariance is a diagonal matrix, we do consider for temporal correlations via a data density
weighting (see Section 2.1).

### 2.2.4 Atmospheric transport

For the synthetic data study only the regional atmospheric model STILT was used to create the
observations with a forward run, and to perform the inversion. This was feasible since the
synthetic $CO_2$ observations are only influenced by fluxes occurring within the DoI, hence global
runs to retrieve boundary conditions at the edge of DoI are not necessary. The transport matrix
for the regional inversions was generated in form of pre-calculated footprints (sensitivities of
atmospheric observations to upstream fluxes) at 0.25 degrees spatial and hourly temporal
resolution for the full year 2007. STILT trajectory ensembles were driven by ECMWF
meteorological fields (Trusilova et al., 2010), and computed for 10 days backwards in time,
ensuring that nearly all trajectories have left the domain of interest.





**2.3 Metrics for performance evaluation**
Following Rödenbeck et al. (2003) we evaluate the goodness of fit for each station (station
specific $\chi^2$). The modeled dry mole fractions should be with 68% probability within the $\pm 1\sigma$
range from the observed mole fractions. This is equivalent to the requirement that the dry mole
fraction part of the cost function defined as the sum of hourly squared differences, divided by the
uncertainty interval and the number of observations $n$ (Eq. 10), should be close to unity.

$$\chi_c^2 = \frac{\sum_t \frac{(\Delta c_t)^2}{\sigma_t^2}}{n} \qquad (10)$$

Another important aspect is the reduced $\chi_r^2$ metric that compares the a-priori model performance
with the specified error structure by dividing the squared residuals of optimized minus observed
dry mole fractions by the squared specified uncertainties. This is also equivalent to two times the
cost function at its minimum divided by the number of degrees of freedom (effective number of
observations) (Thompson et al., 2011):

$$\chi_r^2 = 2\frac{J_{min}}{n} \qquad (11)$$

Again, a correct balance should be close to unity. Smaller values suggest that the model
performance was better than specified in the covariance structure and hence the assumed
uncertainties (denominator) were conservative.
In flux space, we evaluate the inversion performance, by comparing the retrieved flux estimates
against the synthetic fluxes ("true") at different temporal and spatial scales: annually and
monthly integrated fluxes, domain-wide and at country scale. In particular we are interested in
capturing the "true" fluxes down to country scale. For that we assess monthly posterior retrievals
which we compare to reference data ("true" fluxes), country aggregated, using a Taylor diagram.
This diagram provides a concise statistical summary of how well patterns match each other in
terms of their correlation and the ratio of their variances.



## 3 Results

The purpose of the synthetic study is to evaluate the system set-up with a realistic approach. To evaluate the ability of the system to retrieve the synthetic true fluxes we visualize spatially distributed fluxes and we study spatially integrated (domain and national scale) as well as temporally (annual and monthly scale) integrated fluxes.

### 3.1 $CO_2$ mole fractions

A comparison of true and modeled $CO_2$ dry mole fractions from forward runs of the optimized fluxes can reveal the goodness of fit, realized through the optimization process. Such a comparison is presented in Figure 3 for the Schauinsland (SCH) continuous station. Both B1 and S1 inversions significantly reduce the misfit between the synthetic (truth) and the a-priori mole fractions. The RMSD between the prior/posterior from the "true" timeseries for all stations (Table 3) shows an average reduction of around 74% and 76% for the S1 and B1 inversions respectively. Prior correlations (prior vs. true dry mole fractions), have an averaged value of 0.46 which is increased to 0.93 for both inversions. Significant differences between the two inversions were not found apart from a slightly larger decrease of the RMSD for the B1 case. Figure 4 summarizes the capability of the inversions to capture the true signal at each station location in form of a Taylor diagram, indicating that the inversions showed a significant increase of the correlation for all sites. Further the variance of the modeled time-series is significantly closer to the variance of the true signal.

To estimate the goodness of fit we consider the station specific $\chi_c^2$ values (Eq. 10), using here 7-day aggregated residuals instead of hourly to match the temporal scale of one week of the observation error. Values smaller than 1 are found for most of the stations with a mean value of 0.28 and 0.32 for the B1 and S1 cases respectively, suggesting a good fitting performance for all stations and for both inversions. The results are comparable with those found in the Rödenbeck et al. (2003) study. The reduced chi-squared (Eq. 11) was found to be 0.21 for both cases, indicating that the error variance is overestimated making the error assumption rather conservative.



## 3.2 Flux estimates and uncertainties

The spatial distributions of the annual biosphere-atmosphere exchange fluxes for the prior, the known truth, and the posterior cases are presented in Figure 5. Note that annual fluxes between the two biosphere models used for prior fluxes and true fluxes are substantially different. The inversion significantly adjusts the spatial flux distribution mainly in central Europe, where a denser atmospheric network exists. The absolute annual mean difference in fluxes (|mean(true − prior)| and |mean(true − posterior)|) is greatly reduced from 70.8 $gCm^{-2}y^{-1}$ to 14.7 $gCm^{-2}y^{-1}$ and 24.6 $gCm^{-2}y^{-1}$ for the B1 and S1 inversions respectively. Detailed patterns, however, are not well reproduced: the fraction of explained spatial variance in the true fluxes (measures as squared Pearson correlation coefficient) decreases from the prior (0.17) to the posterior (0.07 and 0.06 for the cases B1 and S1, respectively). When evaluating this at monthly scales, the fraction of explained spatial variance increases in the posterior estimates compared to the prior for winter months from around 0-15% to about 15-50%, while during the growing season typically a decrease from around 10-35% to about 0-34% is found. The accumulated footprint of the atmospheric network is shown in Figure 6, clearly indicating the strongest constraint on fluxes in central Europe. Interestingly both error structures from S1 and B1 inversions produce posterior fluxes that have approximately the same spatial distribution. When separating the spatiotemporal component from the bias component (in S1 case) we can identify differences between the two inversions. Significant deviations of the spatial flux distribution between the spatiotemporal components were found: The spatiotemporal component in the S1 case has a domain wide annual flux correction of 0.39 $GtC\ y^{-1}$ (prior – posterior) while the corresponding term in the B1 case has a correction of 0.78 $GtC\ y^{-1}$. Nevertheless standard deviations of the corrections with respect to the true spatial flux distribution (true – posterior) found to have no significant difference ($6.88*10^{-5}$ and $7.38*10^{-5}$ $GtC\ y^{-1}cell^{-1}$ for S1 and B1 respectively). We do not observe any strong correction in the south-eastern part of Europe as it cannot be "seen" from the atmospheric network due to the distance to the observing sites and the prevailing westerly winds. This could also be inferred from the flux innovation plots (see Figure 5) defined as the difference between prior and posterior fluxes. Only very small or even no corrections occurred in this area.

We are specifically interested in the ability of the inversion system to capture integrated fluxes over time and space. Figure 7 shows an overview of the domain-integrated fluxes at a monthly





and annual scale. Despite the remarkably larger a-priori (VPRM) sink compared to the synthetic truth (GBIOME-BGCv1) during the growing season, both inversions, with and without the bias term, produce posterior flux estimates that fully capture the "true" monthly and annually integrated fluxes. While the monthly posterior estimates give no clear evidence on which inversion performs better, retrievals at annual scale slightly favor the inversion without the bias term (B1 case). A difference was observed in the prior uncertainties between the two inversions. While both were scaled to have the same prior annual uncertainty, the B1 inversion has systematically larger prior monthly uncertainties than the S1 as a result of the inflated spatiotemporal component of the prior error covariance. Posterior uncertainties were found to be similar, and include or are close to including (S1 case) the true flux estimates. The uncertainty reduction for annually and domain-wide integrated fluxes, defined as the difference between prior and posterior uncertainties normalized by the prior uncertainty, was found to be 73% and 69% for the S1 and B1 respectively.

In order to assess how well the posterior estimates agree with the true fluxes, root mean square difference (RMSD) between true and posterior monthly integrated gridded fluxes were computed (Table 4). Both inversions B1 and S1 show a similar reduction in the RMSD values compared to the prior. The same picture emerges for the annually integrated fluxes.

Of particular interest is the performance of the system at regional scale, specifically at national level. Figure 8 shows monthly fluxes for selected European countries, including the prior, true and posterior estimates with the corresponding uncertainties. Both error structures show a similar performance. Despite the large prior misfit, the system succeeded in retrieving monthly fluxes at country level. Better constrained regions mainly located in central Europe show the ability to broadly capture the temporal flux variation at monthly scale. Figure 9 summarizes in a Taylor diagram the inversion performance for each EU-27 country, showing the improvement of monthly and country aggregated fluxes (perfect match would be if the head of the arrow coincides with the reference point marked as green bullet). It is worth mentioning that also for regions that are less constrained by the network, such as Great Britain, Spain, Poland and Romania, the inversions still improved the posterior estimates compared to the prior estimates (see also Fig. 9).





### 3.3 Evaluation with synthetic eddy covariance data

In order to investigate the potential of using eddy covariance measurements for evaluating the retrieved $CO_2$ fluxes, monthly fluxes from the prior (VPRM), the truth (GBIOME-BGCv1), and the posterior for cases B1 and S1 were extracted at the grid cell locations where eddy covariance stations exist, using the same 53 sites as in Kountouris et al. (2015). The corresponding fluxes were then aggregated over all sites, using a weight that compensates for the asymmetry between number of flux towers for specific vegetation types and the fraction of land area covered by the specific vegetation type. Prior fluxes show a systematically larger uptake compared to the truth, predominantly during the growing season with maximum differences of 0.8 $gCm^2day^{-1}$ (Figure 10). Posterior estimates for both cases captured the magnitude of the true fluxes, with maximum differences of around 0.3 $gCm^2day^{-1}$ during June/July. A significantly larger correction is apparent during spring and summer compared to winter and fall. The very close correspondence of these results with those shown in Figure **7** for the domain-wide monthly flux budget clearly shows that eddy covariance measurements can principally be used for validation of the inverse estimates at monthly timescales.

## 4 Discussion

### 4.1 Performance in flux space

Results from the synthetic experiment showed the strengths but also the weaknesses of the system to retrieve the "true" spatial flux distribution. Although the error structure applied to this experiment was statistically coherent with the mismatch between prior and true fluxes, we note a limited ability of the current atmospheric network to retrieve fluxes at local scales. For coarser spatial scales (country level) the carbon budget estimates in the synthetic inversion showed a quite good performance at monthly and annual temporal scales. Further we observed an average reduction of the monthly uncertainties of 65% for the B1 case, and 64% for the S1 case. In

combination with the fact that the flux estimates reproduce the "truth" within the posterior
uncertainties, this gives us confidence in the accuracy of our estimates.
Prior error correlation in time and space limits the scale, at which information can be retrieved
from the inversion. The spatial correlation of several hundred kilometers implies that fluxes at
scales smaller than this cannot be significantly improved by the inversion, as the results clearly
showed. To assess this more quantitatively, the spatial correlation between a priori or retrieved
and true monthly fluxes is calculated for different spatial aggregation scales (starting at 0.25
degree, fluxes were aggregated to 0.5, and then in 1-degree steps up to 8 degree). Results shown
in Fig. 11 a) indicate a nearly continuous increase of the spatial correlation of prior and posterior
fluxes with increasing aggregation scale. The additional explained variance brought about by the
inversion, i.e. the difference between posterior (red/blue line) and prior (grey line) flux
correlation (r-square) with the truth, starts at low values around 0.1, and reaches values around
0.2 for scales larger or equal 2 degrees. Similarly, the spatial correlation between a priori and
true fluxes for a given spatial aggregation of 2 degrees, but for different temporal aggregation
scales ranging from 1 day to 128 days (Fig. 11 b) shows a continuous increase from about 0.23 to
0.42 (r-square), while the spatial correlation between retrieved and true fluxes only varies
slightly between 0.4 and 0.53 (Fig. 11 b), red and blue lines). Here, the additional spatial
variance explained by the retrieved fluxes is largest at around monthly time scales (differences
between prior and posterior r-square around 0.2), while at seasonal scales this additional
explained variance is only around 0.1. Overall, this analysis confirms that there are preferred
spatial and temporal scales at which the inversion retrieves the flux distribution best and where
thus most information is gained. This is not dependent on whether or not a bias term is included
in the state vector, as results for case B1 and S1 do not differ in this regard. It is important to
realize that all other scales, at which the inversion does not provide much information, need to be
properly represented by the a priori flux distribution. Thus the a priori fluxes need to be realistic
at short spatial scales below about 200 km, at seasonal temporal scales, and of course at hourly
time scales which are not retrieved by the inversion.
The annual spatial flux distribution of the B1 and S1 cases was found to be quite similar,
indicating that inflating the uncertainty by a factor of 1.5 (B1 case, see also 2.2.1 section) or
adding a bias component to compensate the inflation (S1 case) lead to a similar flux constraint.



This could be explained due to the long correlation length (566 km) which drastically reduces the
effective number of degrees of freedom, forcing the fluxes to be smoothly corrected, regardless
the use of the bias component.
**4.2 Performance in observation space**
The high RMSD reduction in combination with the high correlation values and the captured
variability between posterior and true dry mole fractions in the synthetic experiment suggest a
good performance of the inversion system to retrieve the "true" mixing ratios. Nevertheless this
is not surprising, as the atmospheric data are "fitted" by the inversion, and furthermore the
forward and the inverse runs used identical transport, without any impact from imperfections in
transport simulations.
The uncertainties in the flux space are statistically consistent with the model-model flux
mismatch. However the reduced $\chi_r^2$ values obtained from the inversions were rather small
(around 0.21). This indicates that overall conservative uncertainties were assumed, and the small
$\chi_r^2$ values are a result from the assumed uncertainties in the observation space. Indeed
uncertainties in the observation space include also transport uncertainties; however, given that
the same transport is used to create synthetic observations and to perform the inversion, there is
no actual model-data mismatch related to transport uncertainties, and so the assumed
uncertainties are overestimated.
**5 Conclusions**
This paper describes the setup and the implementation of prior uncertainties as derived from
model-eddy covariance data comparisons into an atmospheric $CO_2$ inversion. The inversion
system assimilates hourly dry air mole fractions from 16 ground stations to optimize 3-hourly
NEE fluxes for the study year 2007. Two different error structures were introduced to describe
the prior uncertainty by either inflating the error or by adding an additional degree of freedom





allowing for a long term bias. The need of this error inflation comes from the fact that the spatiotemporal model - data error structure alone underestimates prior uncertainties typically assumed for inversion systems at continental/annual scale. In this study we evaluate the Jena inversion system by performing a synthetic experiment and expanding the evaluation also to the retrieved fluxes, whilst only the observation space was evaluated in Trusilova et al. (2010). Further we assess the impact when adding a bias term in the flux error structure. This study is a preparatory step to retrieving European biogenic fluxes using a data driven error structure consistent with model-flux data mismatches, which is described in the companion paper (Kountouris et al. 2016).

Significant flux corrections and error reductions were found for larger aggregated regions (i.e. domain-wide and countries), giving us confidence on the reliability of the results for a real data inversion. We found a similar performance for both error structures. A more detailed analysis of the spatial and temporal scales, at which the inversion provides a significant gain in information on the distribution of fluxes, clearly confirms that a) fluxes at spatial scales much smaller than the spatial correlation length used for the a prior uncertainty cannot be retrieved; b) the inversion performs best at temporal scales around monthly, and c) especially the small spatial scales need to be realistically represented in the a priori fluxes.

**Acknowledgments**

This work contributed to the European Community's Seventh Framework Program (FP7) project ICOS-INWIRE, funded under grant agreement no. 313169. The authors would also like to thank the Deutsches Klimarechenzentrum (DKRZ) for using the high performance computing facilities. This publication is an outcome of the International Space Science Institute (ISSI) Working Group on "Carbon Cycle Data Assimilation: How to consistently assimilate multiple data streams publication is an outcome of the International Space Science Institute (ISSI) Working Group on "Carbon Cycle Data Assimilation: How to consistently assimilate multiple data streams.





**Appendix**
The exponentially decaying temporal autocorrelations is a feature newly implemented into the
Jena Inversion System. Temporal correlations are not directly defined as off-diagonal elements
in the a-priori error covariance, as the latter does not appear explicitly in the inversion. Rather,
the inversion system involves time series filtering in terms of weighted Fourier expansions. More
specifically the columns of matrix $G_{tcor}$ contain Fourier modes, weighted according to the
frequency spectrum that corresponds to the desired autocorrelation function. The reader is
referred to Rödenbeck (2005) for more information. Following Rödenbeck (2005) we define the
following spectral weight w:
$$w = \frac{v_{low}}{\sqrt{v_{low}^2 + (2\pi v)^2}}$$   A1
where $v_{low}$ is the characteristic frequency. The characteristic frequency $v_{low}$ can be calculated
from the desired temporal autocorrelation time (30 days) of the exponential decay and is
expressed in years:
$v_{low} = 1/(1/12)$ where 1/12 is the autocorrelation time in years. Hence the characteristic frequency
corresponding to a monthly autocorrelation is 12.
To test numerically whether the implemented autocorrelation decay shape approximates an
exponential decay, an error realization of the characteristic frequency was added to the prior
fluxes, and the autocorrelation function as described in Kountouris et al. (2015) was calculated
numerically simultaneously for the flux time series of all grid cells. Then an exponentially
decaying function was fitted (Fig. A1) to derive the autocorrelation scale for the corresponding
frequency. The resulting autocorrelation shape indeed approximates very well an exponential
decay, with an e-folding time of precisely 30 days. The tight confidence bounds of the fitted
parameter (29.3 and 30.6 days within 95 % confidence interval), in combination with the small
residual sum-of-squares (0.14) suggests a very good approximation of the exponential decay.





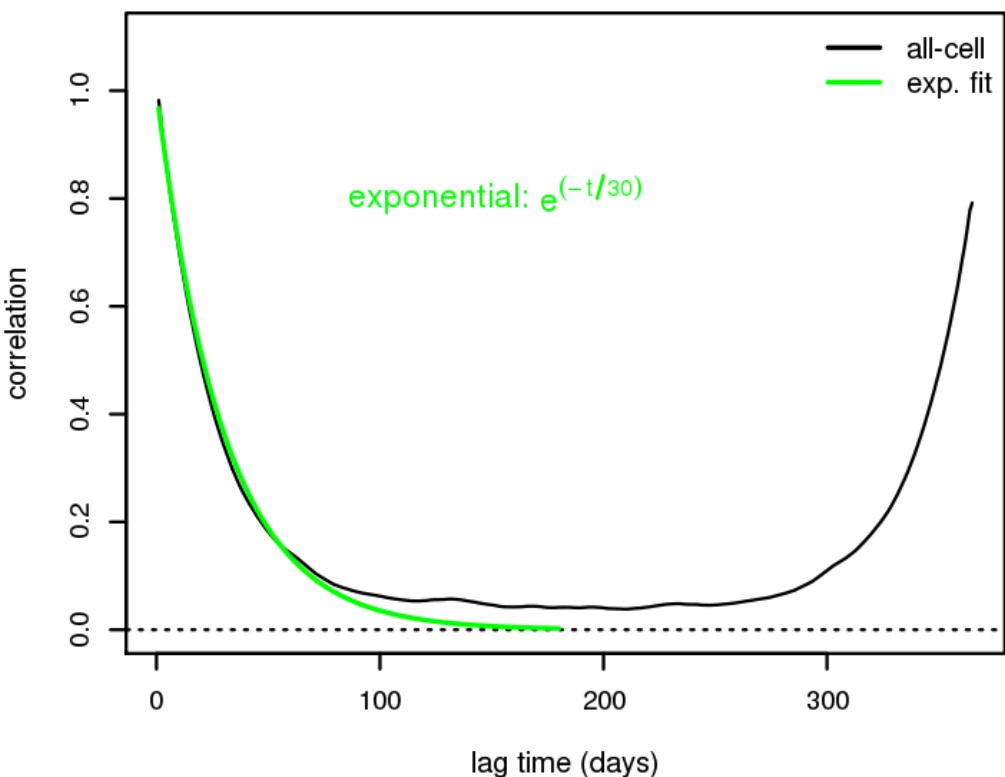

2    Figure A1: Autocorrelation function for a characteristic frequency of the exponential filter. The

3    autocorrelation is calculated simultaneously for all the domain grid cells. The numerical

4    realization of the autocorrelation does not decay to zero because of the flux seasonality.



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



1    Table 1. Optimized VPRM parameters $SW_0$, $\lambda_{SW}$, $\alpha$, $\beta$ for different vegetation classes[a]

|  | $SW_0$ | $\lambda_{SW}$ | $\alpha$ | $\beta$ |
|---|---|---|---|---|
| Evergreen forest | 275 | 0.226 | 0.288 | -1.10 |
| Deciduous forest | 254 | 0.215 | 0.181 | 0.84 |
| Mixed forest | 446 | 0.163 | 0.244 | -0.49 |
| Open shrub | 70 | 0.293 | 0.055 | -0.12 |
| Crop | 1132 | 0.086 | 0.092 | 0.29 |
| Grass | 528 | 0.119 | 0.125 | 0.017 |

2    [a]Units are as follows: $SW_0$ : $W\ m^{-2}$; $\lambda_{SW}$: $\mu mole\ CO_2\ m^{-2}s^{-1}\ /\ (W\ m^{-2})$; $\alpha$: $\mu mole\ CO_2\ m^{-2}s^{-1}\ /\ {}^0C$;

3    $\beta$: ($\mu mole\ CO_2\ m^{-2}s^{-1}$).



1  Table 2. Information on the stations used for the regional inversions. Same network applied for

2  the synthetic, and the real data inversions in Kountouris et al. (2016). In first column the term

3  "type" stands for continuous (C) or flask (F) data.

| Site Code / type | Name | Latitude (°) | Longitude (°) | Height (m.a.s.l.) (m) | Measurement height (above ground) (m) | Model height |
|---|---|---|---|---|---|---|
| BAL/F | Baltic Sea, Poland | 55.50 | 16.67 | 8 | 57 | 28 |
| BIK/C | Bialystok, Poland | 53.23 | 23.03 | 183 | 90 | 90 |
| CBW/C | Cabauw, Netherlands | 51.58 | 4.55 | -2 | 200 | 200 |
| CMN/C | Monte Cimone, Italy | 44.18 | 10.7 | 2165 | 12 | 670 |
| HEI/C | Heidelberg, Germany | 49.42 | 8.67 | 116 | 30 | 30 |
| HPB/F | Hohenpeissenberg, Germany | 47.80 | 11.01 | 934 | 50 | 10 |
| HUN/C | Hegyhatsal, Hungary | 46.95 | 16.65 | 248 | 115 | 96 |
| JFJ/C | Jungfraujoch, Switzerland | 46.55 | 7.98 | 3572 | 10 | 720 |
| KAS/C | Kasprowy Wierch | 49.23 | 19.93 | 1987 | 5 | 480 |
| LMU/C | La Muela, Spain | 41.36 | -1.6 | 570 | 79 | 80 |
| MHD/C | Mace Head, Ireland | 53.33 | -9.90 | 25 | 10 | 15 |
| OXK/C | Ochsenkopf, | 50.03 | 11.81 | 1022 | 163 | 163 |





| | | | | | | |
|---|---|---|---|---|---|---|
| | Germany | | | | | |
| PRS/C | Plateau Rosa, Italy | 45.93 | 7.71 | 3480 | - | 500 |
| PUY/C | Puy De Dome, France | 45.77 | 2.97 | 1465 | 10 | 400 |
| SCH/C | Schauinsland, Germany | 47.92 | 7.92 | 1205 | - | 230 |
| WES/C | Westerland, Germany | 54.93 | 8.32 | 12 | - | 15 |





Table 3. RMSD (first column in ppm) and correlation coefficients (second column) between
known truth and prior/posterior $CO_2$ dry mole fractions for daily "daytime" or "nighttime"
averaged values and for each station. The third column shows $\chi^2$, the normalized dry mole
fraction mismatch per degree of freedom for 7-day averaged residuals, as a measure of how well
the data were fitted. The format for each station is as follows: RMSD | $r^2$ | $\chi^2$.

|  | Prior | B1 | S1 |
|---|---|---|---|
| BAL | 4.78 \| 0.07 \| 18.44 | 0.89 \| 0.97 \| 0.48 | 1.02 \| 0.96 \| 0.37 |
| BIK | 5.28 \| 0.43 \| 15.50 | 1.20 \| 0.97 \| 0.18 | 1.29 \| 0.97 \| 0.25 |
| CBW | 8.60 \| 0.04 \| 74.29 | 0.99 \| 0.99 \| 1.31 | 1.06 \| 0.99 \| 1.34 |
| CMN | 2.68 \| 0.33 \| 6.31 | 0.74 \| 0.93 \| 0.08 | 0.78 \| 0.92 \| 0.10 |
| HEI | 11.39 \| 0.37 \| 12.97 | 1.83 \| 0.98 \| 0.36 | 1.84 \| 0.98 \| 0.37 |
| HPB | 7.73 \| 0.35 \| 26.58 | 1.01 \| 0.99 \| 0.21 | 1.19 \| 0.99 \| 0.31 |
| HUN | 6.50 \| 0.63 \| 31.89 | 1.36 \| 0.98 \| 0.21 | 1.46 \| 0.98 \| 0.25 |
| JFJ | 3.12 \| 0.21 \| 3.93 | 1.24 \| 0.86 \| 0.24 | 1.31 \| 0.84 \| 0.27 |
| KAS | 4.00 \| 0.32 \| 10.67 | 0.73 \| 0.98 \| 0.11 | 0.80 \| 0.97 \| 0.15 |
| LMU | 3.42 \| 0.19 \| 6.5 | 0.79 \| 0.95 \| 0.12 | 0.86 \| 0.94 \| 0.16 |
| MHD | 1.53 \| 0.0002 \| 0.83 | 0.65 \| 0.09 \| 0.16 | 0.68 \| 0.06 \| 0.17 |
| OXK | 6.10 \| 0.21 \| 38.50 | 3.35 \| 0.76 \| 0.76 | 3.40 \| 0.75 \| 0.80 |
| PRS | 2.32 \| 0.15 \| 2.46 | 0.70 \| 0.92 \| 0.30 | 0.74 \| 0.91 \| 0.33 |
| PUY | 4.27 \| 0.15 \| 12.06 | 0.68 \| 0.97 \| 0.06 | 0.73 \| 0.15 \| 0.09 |
| SCH | 4.76 \| 0.26 \| 21.17 | 0.90 \| 0.97 \| 0.07 | 0.95 \| 0.97 \| 0.09 |



1    Table 4. Performance of the two error structures expressed as the spatial RMSD of the optimized

2    monthly and annual NEE fluxes compared to the truth for the whole domain in µmole m$^{-2}$ s$^{-1}$.

|  | Annual | JAN | FEB | MAR | APR | MAY | JUN | JUL | AUG | SEP | OCT | NOV | DEC |
|---|---|---|---|---|---|---|---|---|---|---|---|---|---|
| prior | 0.38 | 0.61 | 0.53 | 0.55 | 1.06 | 1.26 | 1.56 | 1.17 | 0.94 | 0.65 | 0.57 | 0.63 | 0.63 |
| B1 | 0.33 | 0.46 | 0.40 | 0.45 | 0.84 | 0.99 | 1.21 | 1.00 | 0.86 | 0.63 | 0.43 | 0.46 | 0.44 |
| S1 | 0.34 | 0.48 | 0.41 | 0.45 | 0.86 | 1.01 | 1.24 | 1.03 | 0.86 | 0.63 | 0.45 | 0.47 | 0.45 |





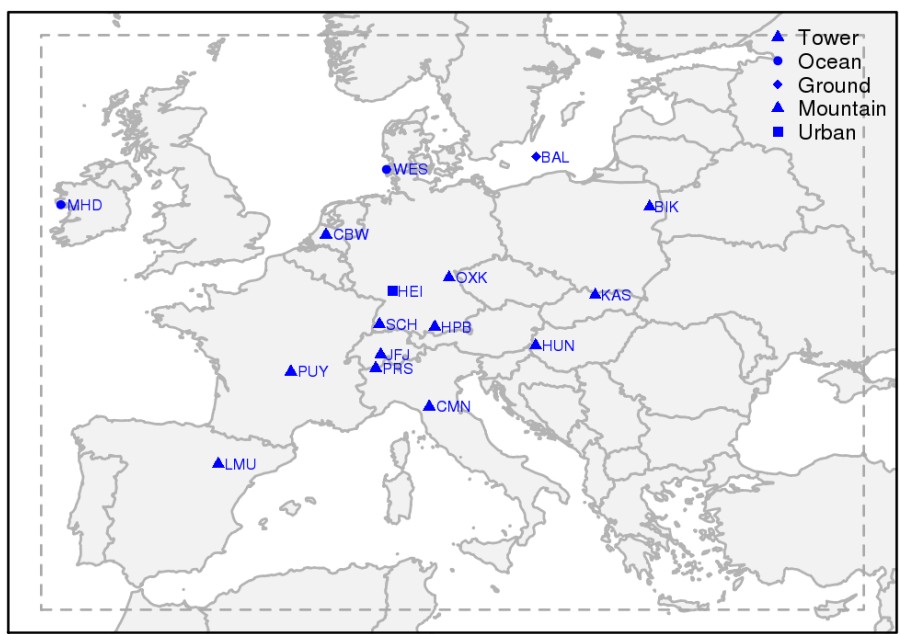

Figure 1. Domain of the inversions (dashed rectangle). Locations of the atmospheric measurement stations are shown with blue marks. Red stars denote the eddy covariance locations used for flux comparisons at grid scale.



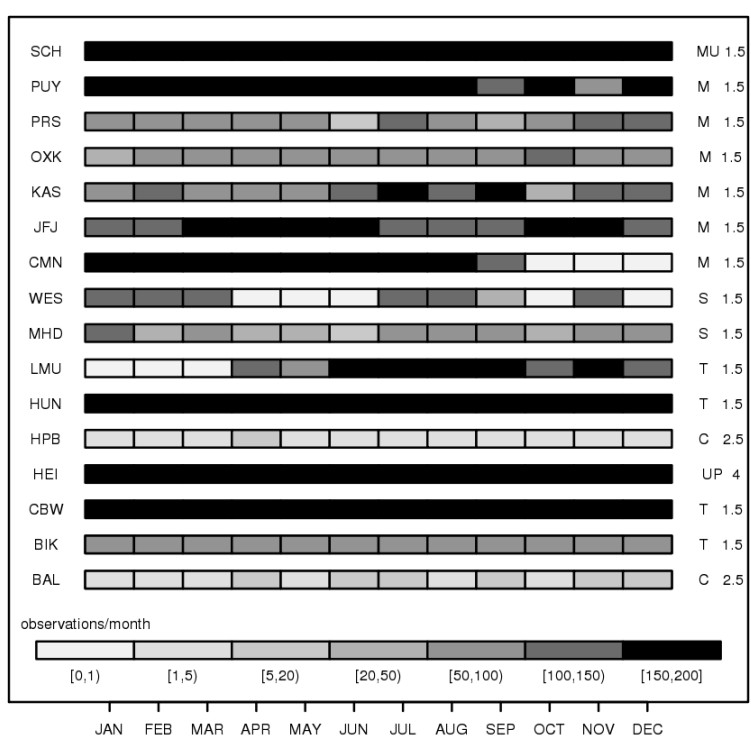

Figure 2. Monthly data coverage plot for the atmospheric stations used in the regional inversions.
Left column shows the code name and the right columns show the station class and the assigned
uncertainty in units of ppm. "C" stands for continental sites near the surface, "T" for continental
tall towers, "S" for stations near shore, "M" for mountain sites, "MU" for mountain sites with
diurnal upslope winds and "UP" for urban pollutant.



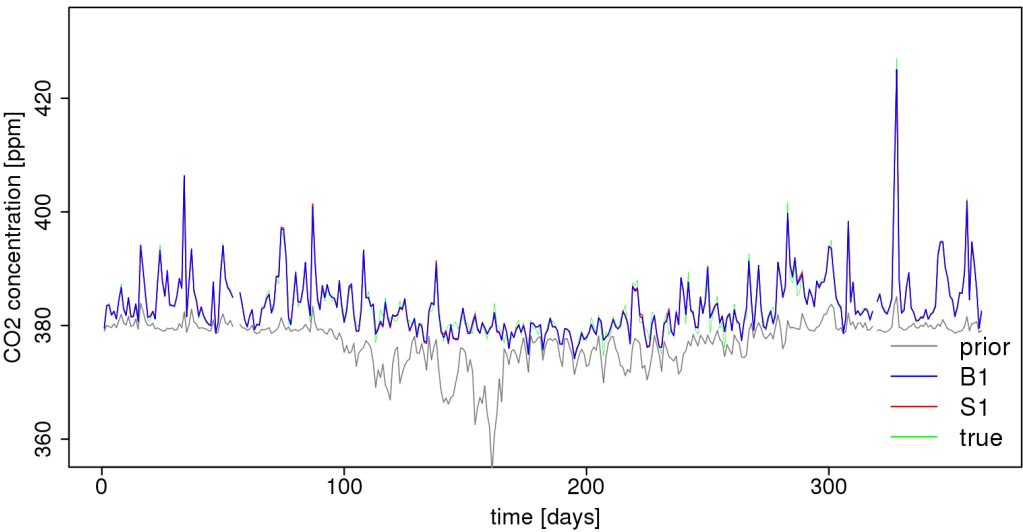

2    Figure 3. Daily nighttime (23:00-4:00 UTC) averages for prior, true, and posterior $CO_2$ dry mole
3    fraction time series for the mountain site Schauinsland. Time starts at $1^{st}$ January 2007.





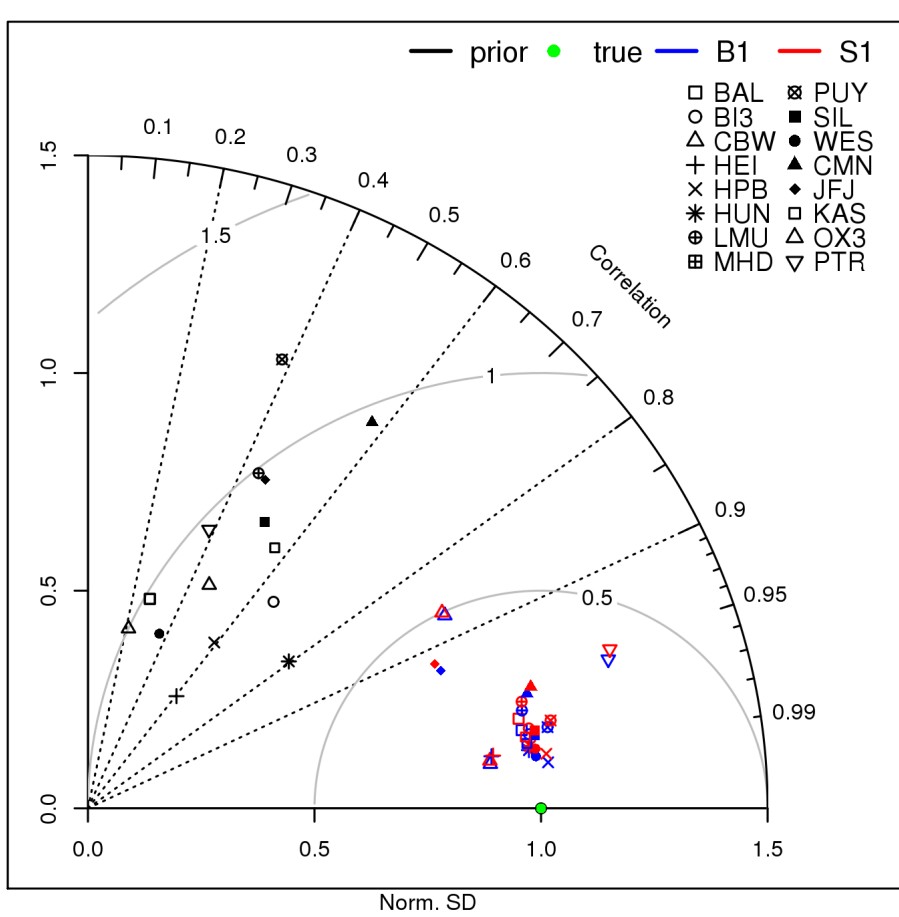

Figure 4. Taylor diagram for modeled and measured time-series of $CO_2$ dry mole fractions. Prior
(black), true (green, the perfect match of modeled and true time-series) and the different
inversion cases (R0 blue; R1 red) are displayed. Different symbols denote different atmospheric
stations. The normalized SD was calculated as the ration of the SD of the modeled time-series to
the SD of observations.





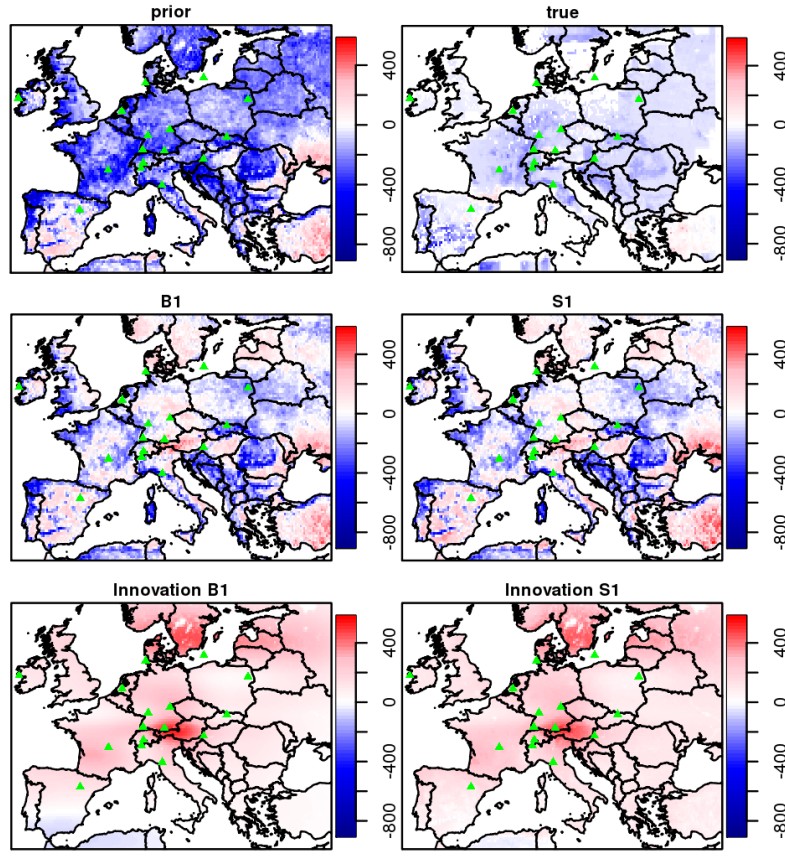

2    Figure 5. Annual spatial distribution for the prior, true, and posterior biogenic flux estimates for
3    the two synthetic inversions S1 and B1 (top two rows), and flux innovation defined as the
4    difference posterior - prior (bottom row). Fluxes are given in units of $gCy^{-1}m^{-2}$.





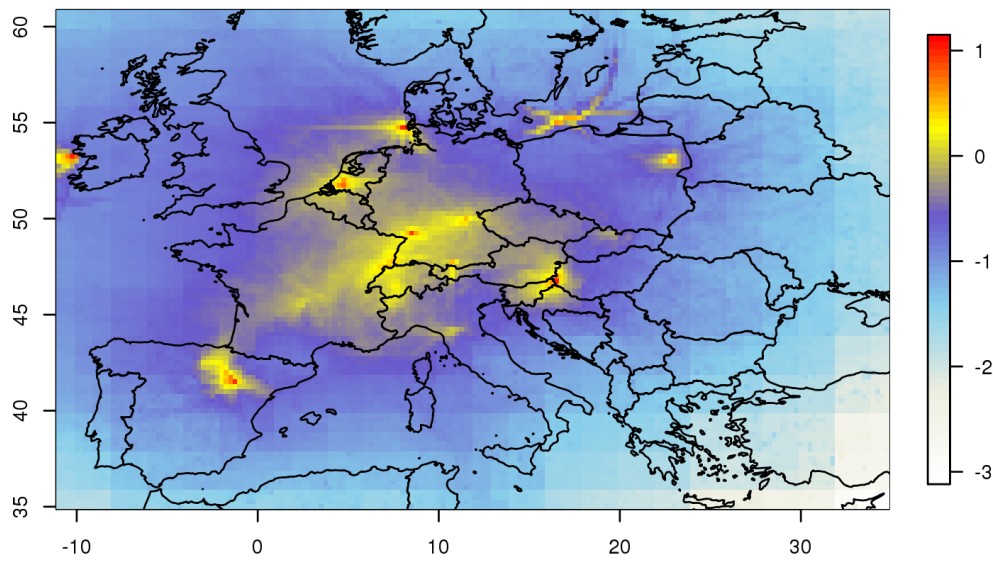

2  Figure 6. Annual integrated influence for 2007 of the current atmospheric network. Footprint
3  influence is presented in a logarithmic scale and units are in $\log_{10}[\mathrm{ppm}/(\mu\mathrm{mol/m^2/s})]$



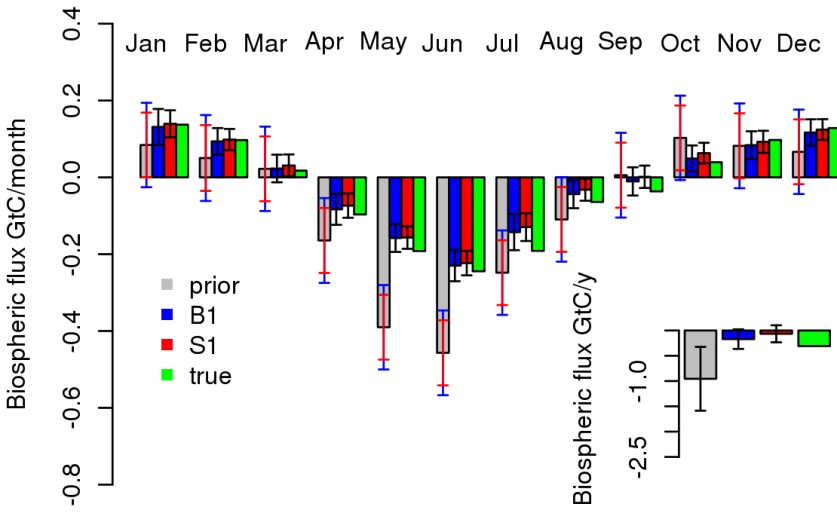

Figure 7. Monthly and annual carbon flux budget, integrated over the European domain. Note
that both inversions share the same annual prior uncertainty but monthly uncertainties differ.
Blue and red error bars denote the prior uncertainty for the B1 and S1 scenarios respectively.



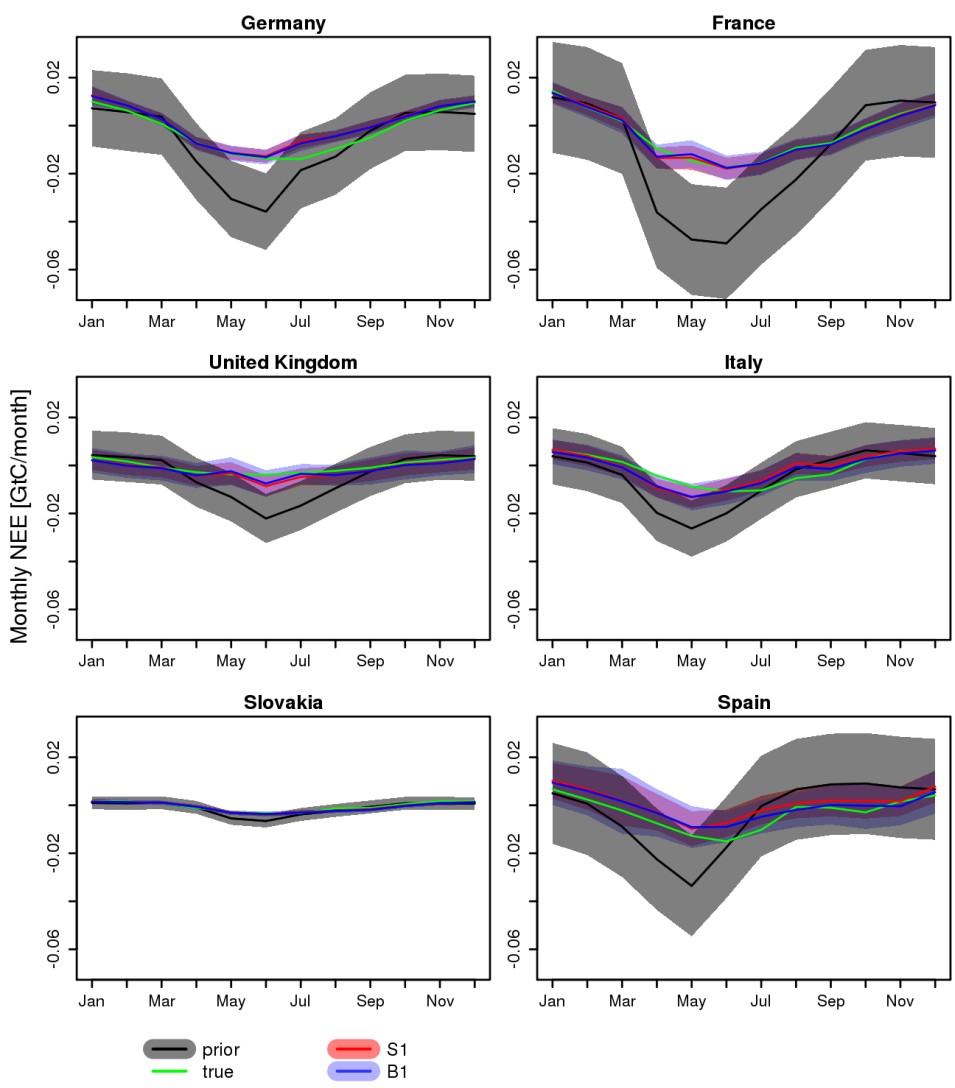

Figure 8. Temporal evolution of monthly NEE for selected European countries for the synthetic
data inversion.



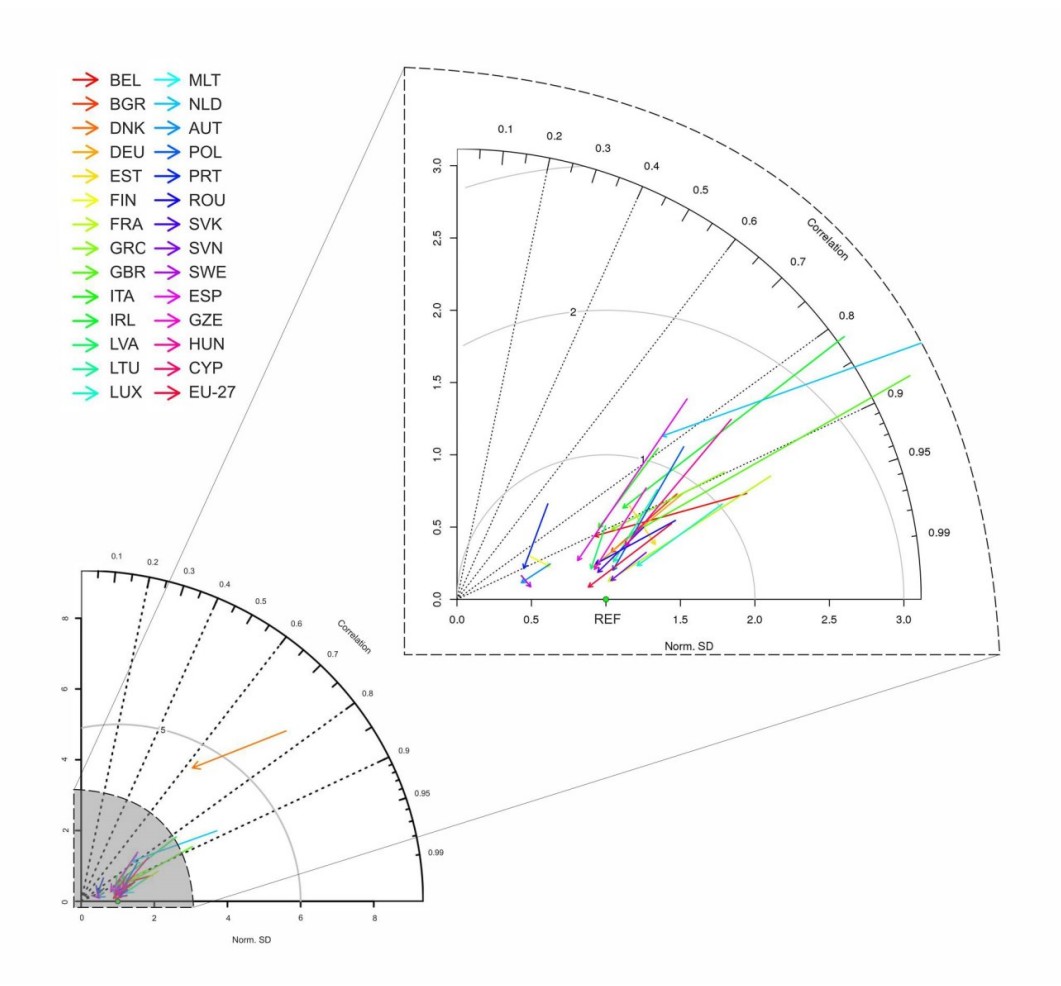

Figure 9. Overview of the model performance summarized in a Taylor diagram. Posterior and prior monthly and country scale aggregated biospheric fluxes are compared against the reference fluxes ("true"). Each line corresponds to a different country. The starting point of each arrow shows prior/reference comparison and the ending point the posterior/reference comparison. Ideally the ending point should coincide with the green point which represents the reference model.





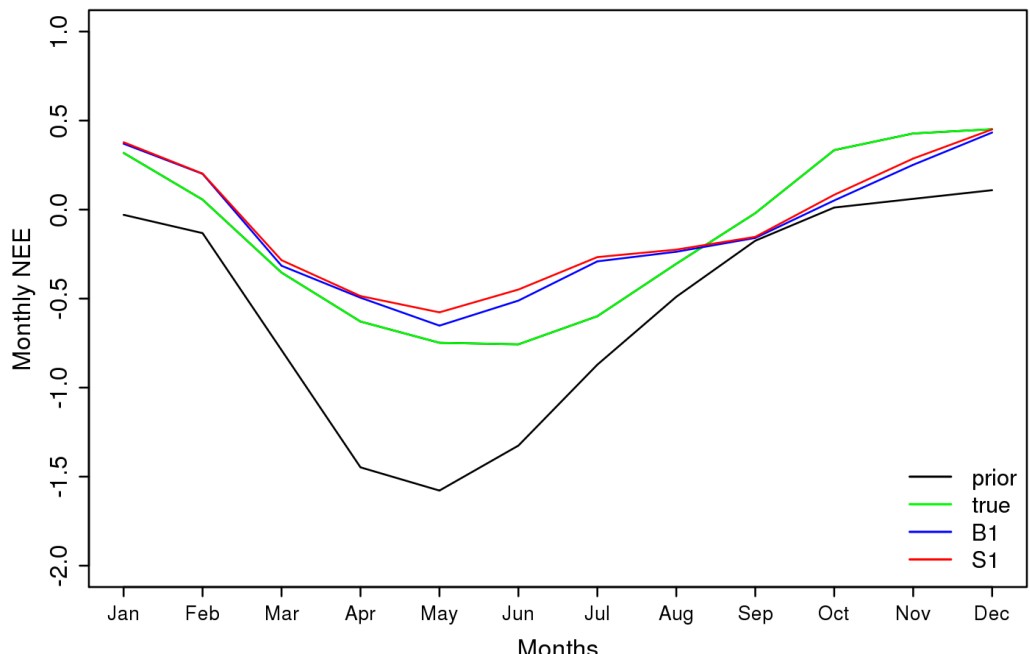

Figure 10. Mean monthly NEE averaged over the 53 different eddy covariance site locations as reported in Kountouris et al. (2015). A priori (black), true (green), and posterior fluxes for scenarios B1 (blue) and S1 (red) are shown. Units are in $gCm^{-2}day^{-1}$.



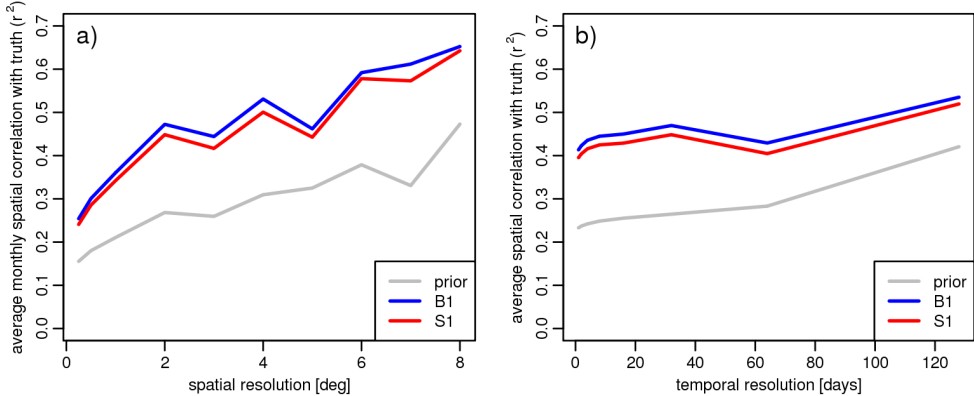

Figure 11. a): Mean spatial correlation of monthly fluxes with true fluxes as function of spatial
flux aggregation scale for prior fluxes (grey), and for posterior fluxes from scenarios B1 (blue)
and S1 (red). b): Mean spatial correlation of fluxes with true fluxes at 2 deg. spatial resolution as
function of temporal flux aggregation scale for prior fluxes (grey), and for posterior fluxes from
scenarios B1 (blue) and S1 (red).

