# Peer review of "Technical Note: Atmospheric CO2 inversions at the mesoscale"

_Atmospheric Chemistry and Physics, 2016_

## Referee Comment (RC1) · Anonymous Referee #3 · 25 Oct 2016

This paper presents a method for estimating a priori flux uncertainties in carbon dioxide inversion systems. The system is then evaluated using synthetic atmospheric data. Whilst the paper is generally well written, I was left wondering what we've really learnt from a study such as this. At present, the abstract and conclusions largely focus on the outcome of the synthetic data inversion, which I don't believe represent a major innovation, or provide a framework that could readily be used in other work (see below). Perhaps the paper can be re-focused on elements that the authors feel represent a true advance, that could be applied beyond the inversion system described. Alternatively, it appears that the authors have attempted to split this work into two publications: whilst I haven't read the companion paper, I wonder whether the work in this paper is

too incremental to stand on its own, and could instead be folded into the other work (provided the below comments can also be addressed)?

General comments:

1. I'm not convinced that, with a synthetic data experiment such as this, it is possible to show whether a particular prior flux uncertainty covariance is closer to the "truth" than another (aside from demonstrating that one or another was obviously very under- or over-constraining), or, put another way, that one inversion set up would perform better using real world data. The paper describes various metrics of the posterior solution. However, most of these (e.g. RMSE and correlation compared to the known fluxes), simply show that the gradient descent is probably working (i.e. these factors must improve unless there is something obviously wrong with the algorithm). The only metric that might have some ability to demonstrate that the prior uncertainty covariance is appropriate to the real world are the chi-squared tests. However, as the authors note, since this is a synthetic data study, the model is "perfect", so the model-data mismatch will be much smaller than would be achieved in the real world, making this test uninformative for real-world applications.

2. Several relevant papers have not been referenced here. Ganesan et al. (2014) tackle essentially the same problem in a hierarchical Bayesian framework. They show that inclusion of a set of hyper-parameters describing the prior uncertainty covariance necessarily moves the posterior uncertainty closer to the "truth", compared to an inversion without these factors. They were also able to include transport model-data mismatch uncertainties in the inversion. Whilst I don't believe they included a spatial or temporal component in the prior uncertainty covariance, they did explore this in the model-data mismatch, and I don't see why the framework couldn't be extended to do so with the prior (similarly the inclusion of a "bias" hyper-prior would also be possible). In a related approach, Lunt et al. (2016) included the spatial disaggregation of the flux field (and hence, presumably, the level of spatial correlation in the posterior solution) as an unknown in the inversion. Finally, Zammit-Mangion et al. (2015; 2016) present

a solution to the flux inverse problem in which only the spatial correlation lengths are used a priori, and the inversion is not constrained to a mean flux field. In summary, I think that these papers demonstrate some significant advances in this area in recent years. Ideally, this article would build on these developments, or demonstrate why the advocated approach is preferable. At the very least, these papers should be cited.

3. In Figure 7, it appears that, for several months, the derived fluxes are not between the prior and the "truth". I'm not sure how this could be the case, since the pseudo-data should always pull the solution towards the truth, and the prior should pull towards itself. Therefore, shouldn't our expectation value of the posterior fluxes be somewhere in between? Has some random error been added to the pseudo-data (this should be clarified in Section 2.2.3)? If so, is this feature a product of this particular random realisation of the pseudo-dataset? Therefore, do you need to run an ensemble of inversions to "average out" sampling errors?

Specific comments:

P4, L31: I don't see why model errors will be more easy to define that prior uncertainties? I don't think we have a very good handle on transport model error. Furthermore, this term does not need to be diagonal, as this sentence implies (see references above).

P6, L30: See references above.

P9, L9: Why limit this matrix to being diagonal? As noted on Line 13, the transport model will certainly exhibit temporal and spatial uncertainty correlations.

P11, L5: This equation is not referenced explicitly in the text. What does it show?

P11, L6 – L12: These terms are discussed before being introduced (they refer to an equation in the following subsection). I think the order needs to be changed here.

P11, L19: If I understand this correctly, synthetic eddy covariance (EC) data were extracted at several locations in both models, and these pseudo-fluxes were used to

calculate the spatial and temporal correlation lengths for use in the inversion (please clarify that this is synthetic EC). So essentially, we are using the difference between two models as a proxy for the uncertainty correlation in the real world? I think this is fine. However, two things come to mind: 1) if we were to use "real" eddy covariance data, we would sample very much smaller length scales than the model (i.e. typically <1km, rather than 50km), so I would not expect that the derived correlations would be comparable to the same experiment using real data (as the text seems to indicate on P6); 2) since we're in model world, and in light of point (1), why not use every grid cell to calibrate the correlations? Would this come out as being very different?

P12, L12: The two experiments that are carried out focus on "tuning" the covariance matrix in two ways, so as to match the overall difference between the two models: B1, scale the covariance matrix uniformly; S1 add a bias. What is the reasoning for choosing only these two methods? Couldn't this mismatch be closed in several other ways, e.g. by increasing the correlation lengths or adding a "nugget" term to the diagonal elements, etc.?

P14, L4-L8: Please provide a reference for these choices of data filtering.

P15, L13: I don't think Thompson et al., 2011 is the most appropriate reference here.

P16, L17 – L22: The improved correlation and "variance" is simply a product of the cost function descent. This should be clarified.

P16, L23: Does "chi-squared" show us anything here that we can extend to the real world, given that the model is perfect (see general point 1 above)?

P17, L7: Again, isn't this a trivial result showing that the gradient descent is working?

P18, L11: Probably should be noted that this will largely be determined by the model-measurement mismatch uncertainty covariance, rather than the prior uncertainty.

P19, L15: I think this is a very strong conclusion to draw here. I'd contend that the suitability of EC data for "validation" of inverse model fluxes is dominated by scaling

issues. In this paper, it is assumed that the EC data is representative of 50km^2. In reality, EC data will sample scales that are orders of magnitude smaller.

P20, L1: I think this shows that your inversion algorithm is working, not that you would get any closer to the truth in the real world.

P21, L13: See general point 1.

P22, L11: I don't think we can comment on the reliability of the results of a real world inversion here. A real world inversion will likely be dominated by chemical transport model errors, which are not quantified here.

References:

Ganesan, A. L., Rigby, M., Zammit-Mangion, A., Manning, a. J., Prinn, R. G., Fraser, P. J., Harth, C. M., Kim, K.-R., Krummel, P. B., Li, S., Mühle, J., O'Doherty, S. J., Park, S., Salameh, P. K., Steele, L. P. and Weiss, R. F.: Characterization of uncertainties in atmospheric trace gas inversions using hierarchical Bayesian methods, Atmospheric Chemistry and Physics, 14(8), 3855–3864, doi:10.5194/acp-14-3855-2014, 2014.

Lunt, M. F., Rigby, M., Ganesan, A. L. and Manning, A. J.: Estimation of trace gas fluxes with objectively determined basis functions using reversible-jump Markov chain Monte Carlo, Geoscientific Model Development, 9(9), 3213–3229, doi:10.5194/gmd-9-3213-2016, 2016.

Zammit-Mangion, A., Cressie, N., Ganesan, A. L., O'Doherty, S. and Manning, A. J.: Spatio-temporal bivariate statistical models for atmospheric trace-gas inversion, Chemometrics and Intelligent Laboratory Systems, 149, 227–241, doi:10.1016/j.chemolab.2015.09.006, 2015.

Zammit-Mangion, A., Cressie, N. and Ganesan, A. L.: Non-Gaussian bivariate modelling with application to atmospheric trace-gas inversion, Spatial Statistics, doi:10.1016/j.spasta.2016.06.005, 2016.

---

## Referee Comment (RC2) · Anonymous Referee #2 · 25 Oct 2016

This paper describes calculations of CO2 fluxes for Europe based on inversion from synthetic concentrations. It serves as preparation of a second part where observed concentrations are used. The title announces that "data driven prior uncertainties" will be used. But there is a substantial issue with this. It is important to note that the paper has a precursor in Kountouris et al. (2015) where prior flux errors are estimated based on comparison of model results and real (eddy correlation) flux observations. There, remarkably small flux error correlation lengths of up to 40 km are found (see page 6 line 7 in the present paper). When this is imposed on the prior flux error matrix, this leads to "exceptionally small" (line 9) estimates of the error in the continental integrated prior flux. Apparently, this constitutes a problem: in the end, the authors decide to
use a much larger correlation length (of 566 km on average, see page 12 lines 3-7), which is based on an investigation of model-model residuals (page 11 lines 19-21, and Abstract). Unfortunately, this means that the "data driven prior uncertainties" claim in the title no longer holds. This also undermines the innovative pretention expressed in the title An interesting innovation is the use of an extra "bias" term in the flux, consisting of a "known" spatial flux field multiplied with an unknown time series to be determined by optimal fitting. This avoids the artificial inflation of errors to obtain an acceptable result. Maybe, more could be said about its proposed physical interpretation (which is now indicated very briefly on page 13 in lines 11-12).

In conclusion, the paper represents little real progress (that is not to say that a lot of technical good work was executed to arrive at this stage), in particular with the synthetic inversion results contradicting the title.

Minor comments Page 2, line 5: "it is used in such a way"  $\rightarrow$  "is used" Page 4, lines 12-16: does this involve nonlinearity ? comment on this. Page 5, line 8: delete "zone" ; "later"  $\rightarrow$  "latter" Page 5, line 10: "with"  $\rightarrow$  "for distances up to"? Page 5, lines 16-19: be more specific Page 5, lines 26-end: this is somewhat difficult to follow. Page 6, line 3: "or"  $\rightarrow$  "respectively" ? Page 6, line 16: "for"  $\rightarrow$  "integrated over" Page 6, line 17: "Although is"  $\rightarrow$  "Although it is". Page 7, line 7: "term referred to a bias term"  $\rightarrow$  "term to reflect the bias" ? Page 7, line 9: "between" : another word is needed here. Page 7, line 27: "conclusions are following in Section 4": these are presently in Section 5. Page 8, line 17: "cini is the initial concentration": is this correct ? With f = 0, cmod would still evolve in time. Page 9, line 6: "constrain"  $\rightarrow$  "constraint" Page 11, lines 1-4: the wording is a bit confused. Page 11, equation 6: apparently not referred to and of unknown use. Page 11, lines 8-12: this is an errant block, it should come later. Page 11, line 12: delete "et al." Page 11, lines 16-21: there is a difference in method here: Kountouris et al. (2015) used model-data instead of model-model comparison. And the resulting correlation lengths are also very different, which should be indicated. Page 11, line 23: "ensuring similarity": same remark. Page 12, lines

**ACPD**
12-13: Not sure if the acronyms "B1" and "S1" would be the best choice, one might think of more telling names. Page 12, line 28: "and unit variance": this pertains not to the adjustable term but to the p-coefficients. Page 13, line 4: "which they a-priori have, a"  $\rightarrow$  "which a priori have a" Page 13, line 6: "derived"  $\rightarrow$  "expressed" (nothing is said yet about how values are derived) General about section 2.2.2: It remains unclear in the paper how posterior errors and covariances are derived. Page 13, line 21: "use a different biosphere model": add eventually references to literature where the same is done, like in the previous sentences. Page 14, line 3: "table 2": and figure 1. Page 14, line 24: "Dol": explain that this means domain of interest. Section 2.3: a separate subsection may be superfluous, instead the content could be built in within the results section. Page 15, line 9 and 10: Unclear sentence. "a-priori" in line 9 and "optimized" in line 10 seem to contradict each other. Page 17, line 6: "central Europe": also south Scandinavia Page 17, line 10: "measures"  $\rightarrow$  "measured". Page 17, lines 11-12: is this shown anywhere in the paper ? Page 17, line 24: "found"  $\rightarrow$  "was found". Page 18, line 24: inversion performance: for which of the two inversions ? see also guestion at figure 9. Page 19, line 10: "Figure": Figure 10. Page 19, line 28: "65 %", "64 %": where is this 14: "years": reciprocal years. Figure 4: "R0", "R1": wrong acronyms. "ration"  $\rightarrow$  "ratio". With which time base were the results obtained ? Figure 5: "gCy-1m-2": usually this is written as "gCm-2yr-1". Figure 9: colors will be often indiscernible in practice (maybe no problem !); why is one arrow seen when there are two ways to calculate a posterior ?

---

## Referee Comment (RC3) · Anonymous Referee #1 · 11 Nov 2016

General comments: The paper is quite well written. The inversion system seemed to do its job pulling the posterior fluxes toward the true flux. However, how accurate is the "true" flux? Perhaps, the author can add more information on how reliable the "truth" was.

Specific comment: P12, L14-15: what was the purpose of setting the bias term according to the annually averaged VPRM respiration only? P19, L10: Specify the Figure number. P19, L10 and 12: gCm2y-1 $\Rightarrow$ gCm-2y-1 P34, Table 2: At some sites, the model and the measurement heights are significantly different. What was the reason for that? And at some sites, the measurement heights were not specified ("-"). P36, Table 3: I noticed that the statistical values of B1 and S1 are quite close except at PUY

site r2 (0.97 and 0.15 for B1 and S1, respectively). I am wondering what happened there. P40, Figure 3: I cannot clearly see the "true" flux line (light green). Perhaps, make it more visible. P42, L4: Usually see gCm-2y-1 instead of gCy-1m-2 P44, Figure 7: Why in many months posterior flux values are not in between prior and true fluxes?

---

## Author Comment (AC1) · 30 Jan 2017

[revised manuscript text omitted]

In the current study we do not excessively assess the transport error but it is rather included as diagonal elements in the measurement error covariance, which is typical in atmospheric inversions. The chi square values confirm that there is no underestimation of the uncertainties. We note though that erroneous flux estimates are likely to be estimated, especially at finer spatial scales where the transport model is not able to resolve the real transport (e.g. individual eddys, complicate terrain etc). However, for coarser spatial scales transport morels perform better, and as long as the fitting performance shows good results, flux estimates should be more reliable.

[revised manuscript text omitted]

**Clarification on how the author response is structured:**

With bold letters we present the comments and questions of the referees. The page and line
numbering is linked to the published paper on the public discussion. Response from the
authors follows a non bold typesetting. Note that page and line numbers are linked to the
corrected and change tracked document.

**Anonymous Referee #1**

We thank the referee for the comments on our manuscript, which helped improving our
study. We hope that our answers and the modifications are satisfactory.

**General comments: The paper is quite well written. The inversion system seemed to**
**do its job pulling the posterior fluxes toward the true flux. However, how accurate is**
**the "true" flux? Perhaps, the author can add more information on how reliable the**
**"truth" was.**

"True" fluxes were created from the GBIOME_BGC biosphere model. This is a terrestrial
ecosystem process model which requires only standard meteorological data like, daily
maximum-minimum temperature, precipitation, incoming shortwave solar radiation, vapor
pressure deficit (VPD), and the day length (DLn). How accurate the modeled fluxes are, is
difficult to say, since this would require a validation against observed fluxes from eddy
covariance stations. The accuracy though of the modeled fluxes is still debatable. We
would like to refer to a study from Friedlingstein et al. (2014). Especially in figure 4 panel
d, we see the large range that terrestrial carbon flux estimates between the different models
exhibit. However, in the current experiment the accuracy of "true" fluxes is not of a
concern. We are interested just to recover the known spatial and temporal flux field using
only atmospheric observations. Accuracy is essential in the part 2 of this study, where real
measurements are used, and the optimized flux field is being validated against real eddy
covariance measurements.

P11, L10 we added:

"The BIOME-BGC model is a terrestrial ecosystem process model which requires only
standard meteorological data like, daily maximum-minimum temperature, precipitation,
incoming shortwave solar radiation, vapor pressure deficit (VPD), and the day length
(DLn). How accurate the modeled fluxes are, is difficult to say, since this would require a
validation against observed fluxes from eddy covariance stations. Nevertheless, biospheric
models still suffer from large uncertainties. The remarkably diverge results between models
confirm how uncertain models are (see Friedlingstein et al. (2014)). However, in the
current experiment the accuracy of the "true" fluxes is not of a concern, since we aim only
to create a synthetic flux field that we perfectly know."

**Specific comment:**

**P12, L14-15: what was the purpose of setting the bias term according to the annually**
**averaged VPRM respiration only?**

The bias shape selection (respiration shape) was preferred over the NEE fluxes, as
otherwise a priori neutral pixels would not receive any flux correction. Further, allowing
the bias to have a spatial shape instead of a flat one, might be sound, since regions with
stronger fluxes might be also more biased." However in part 2 of this study we use
different shapes of the bias term to investigate how the bias spatial structure impacts the
posterior flux estimates.

P13, L18 we added: "The idea behind the implementation of this term is that at large scales
a bias might exists, which can not be captured in the model-data residual autocorrelation
analysis (EC measurements are representative at scales ~ 1 km). This assumption avoids
the artificial inflation of the uncertainty at pixel scale, and restricts the pixel to pixes
corrections to be statistically consistent with the actual error structure. The bias shape
selection (respiration shape) was preferred over the NEE fluxes, as otherwise a priori
neutral pixels would not receive any flux correction. Further, allowing bias to have a spatial
shape might be sound, since regions with stronger fluxes might be also more biased."

**P19, L10: Specify the Figure number.**

It is defined but the number is separated and it is in the next line. We correct though and
now reads "Fig. 10" instead of Figure 10.

**P19, L10 and 12: gCm2y-1 → gCm-2y-1**

P22, L25 and 27 we corrected : "$gCm^{-2}day^{-1}$"

**P34, Table 2: At some sites, the model and the measurement heights are significantly**
**different. What was the reason for that? And at some sites, the measurement heights**
**were not specified ("-").**

Mountain stations are more difficult to represent in models because the orography is
generally not adequately resolved. In STILT we use surface elevation maps from ECMWF
(European Centre for Medium Range Weather Forecasting) with a resolution of 0.25 x 0.25
degrees. As the model orography represents an average over the whole grid cell, it is, in
particular at steep mountain sites, significantly smaller compared to the real orographic
height at the station location. In order to better represent the location of the station in the
large scale flow, usually a model height is assumed that more closely represents the real
height (above sea level) of the measurements. However, using exactly the measurement
height (a.s.l.) in the model would decouple the $CO_2$ signal too strongly from the surface fluxes and hence lead to a systematic underestimation of the surface influence on the concentrations (Geels et al., 2007). We try to find a compromise and adjust the model height (above ground) by half the distance between the model orographic height and the real station height. This assumption is supported by comparisons of modeled and observed diurnal cycles of CO2 concentration at mountain sites.  For sites that a dash (-) appears, means that we have no information about the height

P17 L1 we added: "With respect to the assumed model height, STILT uses surface elevation maps from ECMWF (European Centre for Medium Range Weather Forecasting) with a resolution of 0.25 x 0.25 degrees. As the model orography represents an average over the whole grid cell, it is, in particular at steep mountain sites, significantly smaller compared to the real orographic height at the station location. In order to better represent the location of the station in the large scale flow, usually a model height is assumed that more closely represents the real height (above sea level) of the measurements. However, using exactly the measurement height (a.s.l.) in the model would decouple the $CO_2$ signal too strongly from the surface fluxes and hence lead to a systematic underestimation of the surface influence on the concentrations (Geels et al., 2007). A compromise was reached by adjusting the model height (above ground) by half the distance between the model orographic height and the real station height."

**P36, Table 3: I noticed that the statistical values of B1 and S1 are quite close except at PUY site r2 (0.97 and 0.15 for B1 and S1, respectively). I am wondering what happened there.**

We thank the reviewer for spotting this mistake. We corrected it, the correct value is 0.96 also for the S1 case at this station.

**P40, Figure 3: I cannot clearly see the "true" flux line (light green). Perhaps, make it more visible.**

The reason is that the modeled dry mole fractions (running the model forward using the optimized fluxes) fit almost perfect to the true. Only by plotting the true last would be visible but then, we could not see the posterior estimates. Unfortunately (or maybe not) changing the color would not work.

Figure 3 we added a note in the caption: "Note due to the almost perfect fit posterior and true time-series overlap to each other."

**P42, L4: Usually see gCm-2y-1 instead of gCy-1m-2 P44,**

P46, L4 we corrected : "$gCm^{-2} y^{-1}$"

**Figure 7: Why in many months posterior flux values are not in between prior and true fluxes?**

Please see explanation in response to Referee 3, general comment 3.

**Anonymous Referee #2**

We thank the referee for the comments on our manuscript, which helped improving our study. We hope that our answers and the modifications are satisfactory.

**General comments:**

**This paper describes calculations of CO2 fluxes for Europe based on inversion from synthetic concentrations. It serves as preparation of a second part where observed concentrations are used. The title announces that "data driven prior uncertainties" will be used. But there is a substantial issue with this. It is important to note that the paper has a precursor in Kountouris et al. (2015) where prior flux errors are estimated based on comparison of model results and real (eddy correlation) flux observations. There, remarkably small flux error correlation lengths of up to 40 km are found (see page 6 L 7 in the present paper). When this is imposed on the prior flux error matrix, this leads to "exceptionally small" (L 9) estimates of the error in the continental integrated prior flux. Apparently, this constitutes a problem: in the end, the authors decide to use a much larger correlation length (of 566 km on average, see page 12 Ls 3-7), which is based on an investigation of model-model residuals (page 11 Ls 19-21, and Abstract). Unfortunately, this means that the "data driven prior uncertainties" claim in the title no longer holds. This also undermines the innovative pretention expressed in the title. An interesting innovation is the use of an extra "bias" term in the flux, consisting of a "known" spatial flux field multiplied with an unknown time series to be determined by optimal fitting. This avoids the artificial inflation of errors to obtain an acceptable result. Maybe, more could be said about its proposed physical interpretation (which is now indicated very briefly on page 13 in Ls 11-12).**

The authors would like to clarify the scope of this paper. In part 1 (current paper) of this study, we evaluate the inversion system in a synthetic experiment by using the same methodology as in the real data inversion. In the synthetic experiment, the known truth, as well as the prior are derived from biospheric models. The real error structure for this case clearly can not be derived from the analysis of model-data mismatch as in Kountouris et al. (2015), but instead from the analysis of model-model mismatch, i.e. the mismatch between true fluxes and a priori fluxes in the synthetic experiment. This paper should be considered as an evaluation of the method we follow, to quantitatively characterize the prior error structure, rather than making assumptions or using expert knowledge. Then in part 2 of this study, which uses real data, we do make use of data driven uncertainties, as those investigated in Kountouris et al. (2015).

Regarding the bias term we added more information in the paper.

P13, L18-25: now reads:

"…The idea behind the implementation of this term is that at large scales a bias might exists, which can not be captured in the model-data residual autocorrelation analysis (EC measurements are representative at scales ~ 1 km). This assumption avoids the artificial inflation of the uncertainty at pixel scale, and restricts the pixel to pixel corrections to be statistically consistent with the actual error structure. The bias shape selection (respiration shape) was preferred over the NEE fluxes, as otherwise a priori neutral pixels (with zero NEE) could not be bias corrected. Further, allowing bias to have a spatial shape might be sound, since regions with stronger fluxes might be also more biased."

Minor comments

**P2, L5: "it is used in such a way", "is used"**

P2, L5 we corrected: "…but is used to.."

**P4, Ls 12-16: does this involve nonLinearity ? comment on this.**

Yes, this involves nonlinearity. We have added this in the introduction:

P4, L16 we added: "…instead of the fluxes themselves; this CCDAS approach also allows for nonlinear dependencies of the fluxes on the parameters."

**P5, L8: delete "zone"; "later", "latter"**

P5, L10 we corrected: "…the climate of the latter varies…"

**P5, L10: "with", "for distances up to" ?**

P5, L12 we clarify: "This leads to correlation lengths approximately two times larger compared to the pulse length."

**P5, Ls 16-19: be more specific**

P5, L22- L25 we clarify: ". In particular correlations are applied such that the same ecosystem types in different TransCom regions (basis function regions, see also http://transcom.project.asu.edu/transcom03_protocol_basisMap.php)                    decrease exponentially with distance (L=2000km), and thus assumes a coupling between the behavior of the same ecosystem."

**P6, Ls 26-end:  this is somewhat difficult to follow.**

P6 L2-L7 we clarify: "They simultaneously estimate posterior fluxes as well as parameters controlling the model-data mismatch uncertainty and the prior flux uncertainty, including variance as well as spatial and temporal correlation lengths. Although this approach may be considered as an objective way to infer spatial and temporal correlation lengths, it forces the structural parameters of the error covariance to be statistically consistent with the atmospheric data. In other words, flux parameters are optimized from atmospheric concentration data, and they are forced to have values which can reproduce the atmospheric data."

**P6, L3:  "or", "respectively" ?**

P6, L17 we clarify: "…models and fluxes measured…"

**P6, L16:  "for", "integrated over"**

P6, L30: we corrected: "…NEE fluxes integrated over the""

**P6, L17: "Although is", "Although it is".**

P6, L31: we corrected: "…Although it is…"

**P7, L7: "term referred to a bias term", "term to reflect the bias" ?**

P7, L21: we corrected: "…term to reflect a bias term."

**P7, L9:  "between" :  another word is needed here.**

P7, L23 : we clarify: "…which couples…"

**P7, L27: "conclusions are following in Section 4": these are presently in Section 5.**

P8, L10 we clarify: "…4 and 5, respectively."

**P8, L17: "cini is the initial concentration": is this correct? With f =0, cmod would still evolve in time.**

We assume a well-mixed (uniform) initial concentration at the beginning of the model run, thus it just remains a constant concentration offset throughout the simulation. A well-mixed initial concentration can be assumed because any spatial structure would be lost during the spin-up period anyway. Deviations of the true initial conditions from $c_{ini}$ are taken into account through flux adjustment during the spin-up period.

P9 L4 we added: "The initial concentration assumed to be well mixed and remains constant throughout the simulation. The assumption of the well mixed initial concentration is considered to be valid, since any spatial structure would be lost during the spin-up period."

**P9, L6: "constrain", "constraint"**

P9, L20: we corrected: "…constraint…"

**P11, Ls 1-4: the wording is a bit confused.**

P11, L22 we clarify: "We note that for the synthetic case the last two a priori terms are set to zero. Similarly the deviation term (the data-derived correction to the a-priori fluxes) of the flux model (Eq. 6) consists of the terms referring to NEE, fossil fuel, and ocean fluxes. Here in the synthetic case the last two terms are set to zero (i.e. they are not optimized)."

**P11, equation 6: apparently not referred to and of unknown use.**

P11, L23: we added the equation in the text

**P11, Ls 8-12: this is an errant block, it should come later.**

P12, L4-8: the text has been moved.

**P11, L12: delete "et al."**

P14, L1 we corrected: "…to Rödenbeck (2005)"

**P11, Ls 16-21: there is a difference in method here: Kountouris et al. (2015) used
model-data instead of model-model comparison. And the resulting correlation lengths
are also very different, which should be indicated.**

P12, L18-21 we clarify: "We note that the current study does not directly make use of the
error structure derived in Kountouris et al. (2015), since this is applicable for real data
inversions. Instead we use the same methodology to derive the actual model-model error
structure since here we perform a synthetic data inversion, exploring amongst others the
accuracy of this method."

However, we do not claim that we use model-data error structure in this study. We have
explicitly written in the same paragraph, that the error structure is estimated according to
the method followed by Kountouris et al. (2015) and that the residual autocorrelation
analysis is performed for the model-model residuals.

**P11, L23: "ensuring similarity": same remark.**

Similarity does not mean the same error structure but rather the same methodology used to
characterize the error structure (e.g. which analysis is used, which stations are assumed and
at which locations, at which spatial temporal scales etc).

**P12, Ls 12-13: Not sure if the acronyms "B1" and "S1" would be the best choice, one
might think of more telling names.**

Although the reviewer is right about the names those acronyms serve a distinct role. First
we make sure we are consistent with the acronyms of the second part of this study.
Secondly in the second part, quite a number of inversions are performed. We consider that
this frugal acronym would be the best choice for the reader to distinguish later the different
inversions. A table explaining all the different scenarios is given in the second part of this
work (table 2 Kountouris et al., 2016)

**P12, L28: "and unit variance": this pertains not to the adjustable term but to the p-
coefficients.**

P14, L9 we corrected by deleting the sentence "and unit variance"

**P13, L4: "which they a-priori have, a", "which a priori have a"**

P14, L13: we corrected: "parameters which a priori have a"

**P13, L6: "derived", "expressed" (nothing is said yet about how values are derived)**

P14, L23 we corrected and now it reads "expressed"

**General about section 2.2.2: It remains unclear in the paper how posterior errors and covariances are derived.**

P15, L6-9 we clarified: "Following Rodgers 2000, the posterior flux uncertainties are contained in the covariance matrix of the posterior probability distribution which can be estimated from eq. (10)

$$Q_{f,post} = ((A \cdot F)^T \cdot Q_c^{-1} \cdot (A \cdot F) + Q_{f,pr}^{-1})^{-1} \qquad (10)$$

where $Q_c$ is the measurement error covariance matrix."

**P13, L21: "use a different biosphere model": add eventually references to literature where the same is done, like in the previous sentences.**

P15, L18 we clarify: "Such an approach is also used by Tolk et al. (2011)."

**P14, L3: "table 2": and figure 1.**

P15, L27 we corrected: "table 2 and Fig. 1. "

**P14, L24: "DoI": explain that this means domain of interest.**

P16, L23 we clarify: "…within the Domain of Interest (DoI)"

**Section 2.3: a separate subsection may be superfluous, instead the content could be built in within the results section.**

We modified by deleting the section 2.3. The content in now incorporated into sections 3.1 and 3.2

**P15, L9 and 10: Unclear sentence. "a-priori" in L9 and "optimized" in L10 seem to contradict each other.**

In statistics the reduced chi square is a weighted sum of squared deviations. The nominator consists of the squared differences between the observations (i.e "true" fluxes) and the calculated data (i.e. the optimized fluxes). The denominator represents the input variance (i.e the a priori uncertainties). Hence by definition, this metric evaluates the a priori uncertainties by comparing them to the squared residuals between observed and optimized data.

P19, L19 we clarified: "Another important aspect is the reduced $\chi_r^2$ metric, which we use to assess the model performance. By definition the reduced $\chi_r^2$ can be obtained by dividing the squared residuals of optimized minus observed dry mole fractions by the squared specified uncertainties."

**P17, L6: "central Europe": also south Scandinavia**

P20, L17 we corrected: "in central Europe and in southern Scandinavia,"

**P17, L10: "measures", "measured".**

P20, L22 we corrected: "measured"

**P17, Ls 11-12: is this shown anywhere in the paper?**

We computed and present within the text the squared Pearson correlation coefficients. The authors feel that it would be superfluous to make a plot or a table for those 3 coefficients.

**P17, L24: "found", "was found".**

P21, L5 we corrected: "…(true – posterior) was found to have…"

**P18, L24: inversion performance: for which of the two inversions? see also question at figure 9.**

P22, L9 we clarify: "…performance for the S1 case and for each…"

**P19, L10: "Figure": Figure 10.**

We corrected: "Fig. 10"

**P19, L28: "65 %", "64 %": where is this stated ?**

This is a trivial calculation which we state directly here.

**P20, L9: "nearly continuous", "nearly monotonous" ?**

P24, L2 we corrected: "nearly monotonous"

**P23, L14: "years": reciprocal years.**

The autocorrelation time is given in years, and amounts to 1/12 years. So it is not "reciprocal years".

**Figure 4: "R0", "R1": wrong acronyms. "ration", "ratio". With which time base were the results obtained ?**

Figure 4 : We corrected and clarified: "Taylor diagram for daily averaged modeled and measured time-series (annual basis) of $CO_2$ dry mole fractions. Prior (black), true (green, the perfect match of modeled and true time-series) and the different inversion cases (B1 blue; S1 red) are displayed. Different symbols denote different atmospheric stations. The normalized SD was calculated as the ratio of the SD..."

**Figure 5: "gCy-1m-2 ": usually this is written as "gCm-2yr-1 ".**

We corrected and now reads "$gCm^{-2} y^{-1}$"

**Figure 9: colors will be often indiscernible in practice (maybe no problem !); why is one arrow seen when there are two ways to calculate a posterior?**

This figure is meant to give an overview of the model performance at country scale. For a more detailed and country specific assessment we used figure 8, but extending to all countries might be superfluous. Hence we make use of the tailor diagram (fig. 9) to summarize the results. We use only the S1 case for the sake of simplicity, since another 27 arrows in the plot would make it not readable. Further, as the results suggest, the two methods are very close to each other, hence it would not add any significant information.

**Anonymous Referee #3**

We thank the referee for the comments on our manuscript, which helped improving our study. We hope that our answers and the modifications are satisfactory.

**Introductory remark:**

**Whilst the paper is generally well written, I was left wondering what we've really learnt from a study such as this. At present, the abstract and conclusions largely focus on the outcome of the synthetic data inversion, which I don't believe represent a major innovation, or provide a framework that could readily be used in other work (see below). Perhaps the paper can be re-focused on elements that the authors feel represent a true advance, that could be applied beyond the inversion system described. Alternatively, it appears that the authors have attempted to split this work into two publications: whilst I haven't read the companion paper, I wonder whether the work in this paper is too incremental to stand on its own, and could instead be folded into the other work (provided the below comments can also be addressed)?**

We disagree with the reviewer in that the paper is too incremental to stand on its own. It is correct that the work has been split into two publications, as this would increase the readability of the paper. The current split helps also a reader that is only interested in the methodological part of the study, including the prior error characterization and the inverse system description. A reader that is interested more in the real data results and the regional European carbon budget including series of sensitivity and case runs, can directly refer to the second part, avoiding all the theory and methodology used behind the inversions.

**General comments:**

**1.**

**I'm not convinced that, with a synthetic data experiment such as this, it is possible to show whether a particular prior flux uncertainty covariance is closer to the "truth" than another (aside from demonstrating that one or another was obviously very under- or over-constraining), or, put another way, that one inversion set up would perform better using real world data. The paper describes various metrics of the posterior solution. However, most of these (e.g. RMSE and correlation compared to the known fluxes), simply show that the gradient descent is probably working (i.e. these factors must improve unless there is something obviously wrong with the algorithm). The only metric that might have some ability to demonstrate that the prior uncertainty covariance is appropriate to the real world are the chi-squared tests. However, as the authors note, since this is a synthetic data study, the model is**

**"perfect", so the model-data mismatch will be much smaller than would be achieved in the real world, making this test uninformative for real-world applications.**

The reviewer argues that the metrics of the posterior solution just support the obvious, that the algorithm works properly. We disagree that the solution in the flux space (RMSE and correlation were explicitly stated in the comment) will obviously converge to the synthetic one. The conjugate gradient algorithm optimizes flux scaling parameters by minimizing the model-data mismatch in the concentration space and not in the flux space. Hence, metrics assessing the posterior solution in concentration space, they should indeed improve, and confirm that the algorithm works. However in the flux space quite different flux patterns would lead to almost the same value of the cost function (Kaminski and Heimann 2001). Bayesian inversions set a limit to the flux field, by accounting the a-priori information. Nevertheless, priors do have a Gaussian a-priori probability distribution and they can deviate from their mean value (best guess value). Hence, convergence of the fluxes to the right direction can not be considered as granted. Thorough analysis is needed to explore first, if the fluxes have indeed converged to the "known truth" and secondly, at which spatial/temporal scales can we retrieve the "known truth".

Regarding the second part of the comment, indeed in the current synthetic experiment, transport uncertainties are not included since the same transport model was used for both, the synthetic data creation and the inversion. However we try to keep the experiment as realistic as possible by assuming two totally different biosphere models, to simulate the synthetic observations (BIOME-BGC, process based model), and to provide the prior flux field (VPRM, diagnostic model). Further, we include data gaps in the synthetic mixing ratios in accordance with the gaps appeared in the real observations, for the same time period (see also 2.2.3). The model-data mismatch is calculated for both, the synthetic case and the real data inversion in the companion paper. Whilst the reviewer expects a large difference between those two inversions, this appears not to be the case. Comparing the mixing ratio time-series plots for the Schauinsland station and also the summarized statistics in the Taylor diagrams for both inversions (synthetic and real data), the model-data mismatch is not dramatically different. Posterior mixing ratios from both inversions share same correlations (above 0.9) with the (true respectively pseudo-) observations; furthermore, the normalized standard deviations show a lot of similarity between the pseudo-data inversion and the real-data inversion. We therefore disagree that the current test is uninformative for a real-world application.

**2.**

**Several relevant papers have not been referenced here. Ganesan et al. (2014) tackle essentially the same problem in a hierarchical Bayesian framework. They show that**

**inclusion of a set of hyper-parameters describing the prior uncertainty covariance necessarily moves the posterior uncertainty closer to the "truth", compared to an inversion without these factors. They were also able to include transport model-data mismatch uncertainties in the inversion. Whilst I don't believe they included a spatial or temporal component in the prior uncertainty covariance, they did explore this in the model-data mismatch, and I don't see why the framework couldn't be extended to do so with the prior (similarly the inclusion of a "bias" hyper-prior would also be possible). In a related approach, Lunt et al. (2016) included the spatial disaggregation of the flux field (and hence, presumably, the level of spatial correlation in the posterior solution) as an unknown in the inversion. Finally, Zammit-Mangion et al. (2015; 2016) present a solution to the flux inverse problem in which only the spatial correlation lengths are used a priori, and the inversion is not constrained to a mean flux field. In summary, I think that these papers demonstrate some significant advances in this area in recent years. Ideally, this article would build on these developments, or demonstrate why the advocated approach is preferable. At the very least, these papers should be cited.**

Ganesan et al. (2014) perform a hierarchical Bayesian inversion for a totally different tracer ($SF_6$). $SF_6$ flux information from direct observations is not available (no eddy covariance (EC) measurements), hence describing the prior error from comparisons to flux observations is rather impossible. For that they use atmospheric mixing ratio measurements to derive optimized fluxes and hyper-parameters. The latter is introduced as an uncertainty term optimized by the atmospheric data (as well as the fluxes are). With this term Ganesan et al. claim to obtain better results than with a traditional Bayesian inversions that use expert knowledge to determine prior error structure. We note two things: 1. we do not use expert knowledge for the prior error covariance but instead a fully characterized error structure, using an autocorrelation analysis in flux residuals. 2. We are able to perform this analysis simply because flux data is available through the EC measurements, something that Ganesan et al. (2014) can not use since there is no $SF_6$ flux information. Since we do have spatial and temporal information for $CO_2$ fluxes, and we can directly quantify the prior error structure we do not see the reason to use mixing ratio measurements to indirectly correct posterior flux estimates by a hyper-parameter which is again optimized from the mixing ratio measurements.

Zammit-Mangion et al. (2015; 2016) and Lunt et al. (2016) studies describe $CH_4$ inversions. Again there is no EC flux information available. Further, $CH_4$ fluxes taken from inventories are quite uncertain. For that, they use a spatially invariant prior flux field claiming that the optimization will be predominately data driven. However the covariance function which describes the spatial dependence in the flux field was obtained by a variogram analysis with fluxes derived from different inventories. In our study instead, we look into spatial and temporal autocorrelation patterns of residuals between flux observations (or pseudo observations) and prior fluxes. This is a direct way to fully characterize the prior error structure, as long as the available information exists (i.e EC flux measurements).

We added a reference also to those papers:

P6, L10-15 : "In a similar approach Ganesan et al. (2014) and Lunt et al. (2016), applied a hierarchical Bayesian model using atmospheric concentrations, to estimate both fluxes, and a set of hyper-parameters (e.g. mean and standard deviation of a priori emissions PDF as well as model – measurement standard deviation and autocorrelation scales). In those studies direct flux information for the tracers of interest (sulfur hexafluoride ($SF_6$) and methane ($CH_4$)) is not available."

**3.**

**In Figure 7, it appears that, for several months, the derived fluxes are not between the prior and the "truth". I'm not sure how this could be the case, since the pseudo-data should always pull the solution towards the truth, and the prior should pull towards itself. Therefore, shouldn't our expectation value of the posterior fluxes be somewhere in between? Has some random error been added to the pseudo-data (this should be clarified in Section 2.2.3)? If so, is this feature a product of this particular random realisation of the pseudo-dataset? Therefore, do you need to run an ensemble of inversions to "average out" sampling errors?**

Contrary to intuition, the posterior expectation value of a synthetic experiment does not necessarily need to be in between the prior and the truth. A possible reason to cause the a-posteriori fluxes to fall outside this bracket are a-priori flux correlations. For illustration, let us consider 2 example pixels with a mutual distance within the spatial correlation radius. Assume that pixel 1 is well constrained by the atmospheric data, and that its true flux is smaller than the prior.

Consequently, the flux correction at such a constrained pixel will have a negative sign. In contrast, assume pixel 2 to have a true flux larger than the prior, and to be poorly constrained. Due to the weak data influence (the network sensitivity is uneven), the flux correction at pixel 2 will mainly follow the constrained pixel 1 via the spatial correlation and be negative as well, even if that brings the a-posteriori flux further away from the true flux.

This scenario is consistent with the behavior of the S1 case relative to the B1 case. In S1, a bias term has been assumed, which simultaneously shifts the flux field at all pixels, whether they are well or poorly constrained. This introduces an additional spatial correlation, possibly causing the S1 fluxes to be outside prior and truth more frequently than in the B1 case.

We note that the pseudo-data does not contain any kind of random error realization, therefore an ensemble of inversions is not required. We clarify that also in the paper:

P15, L20: "…We note that the synthetic data were derived without adding error realizations."

**Specific comments:**

**P4, L31: I don't see why model errors will be more easy to define that prior uncertainties? I don't think we have a very good handle on transport model error. Furthermore, this term does not need to be diagonal, as this sentence implies (see references above).**

We clarify that uncertainties in the measurements may be easier to quantify. We added:

P4, L31: "While the measurement uncertainty in the observational constraint is usually defined with the main diagonal of the covariance matrix representing the uncertainty of the observations and the model at a specific time and location, our knowledge for the prior uncertainty is limited, especially regarding temporal and spatial correlations that effectively control the state space."

We agree with the reviewer that the measurement error covariance matrix (includes measurement and transport uncertainties) does not have to be strictly diagonal as correlations are probably present. Whilst we do not explicitly introduce off-diagonal terms in the measurement error covariance matrix, the Jena Carbonscope system implicitly assumes that correlations exist. In fact the system contains the so called "density weighting function" (Rödenbeck, C., 2005). The role of this weighting is to combine flask (~weekly) and continuous (hourly) data with a consistent way, as otherwise the high frequency data would lead to a stronger impact at those particular sites. To avoid that, the density weighting inflates the uncertainty by the square root of the number of the observations at weekly basis. This density weighting plays one more role. It implicitly takes into account correlations in transport uncertainties which might be present. More information can be found in Rödenbeck, C., 2005.

P16, L14 we added: "…transport error correlations might be present. Although we do not explicitly introduce off-diagonal terms in the measurement error covariance matrix, we do consider for temporal correlations via a data density weighting function which inflates the uncertainty. (see Section 2.1 and more information in Rödenbeck, C., 2005)."

**P6, L30: See references above.**

This is not described in the references mentioned by the referee, as those publications do not make use of flux observations to constrain the a priori flux error structure.

**P9, L9: Why limit this matrix to being diagonal? As noted on L 13, the transport model will certainly exhibit temporal and spatial uncertainty correlations.**

As with the comment above, we agree with the reviewer that the transport model will exhibit temporal and spatial error correlations. We have also explained above, the role of the density weighting. The reviewer might have some concerns regarding the concentration mismatch uncertainty, that without considering the correlations we might have underestimate it. At this point, we would like to refer to the reduced chi square values at site level (eq. 11). This metric by definition (posterior mismatch over assumed uncertainties) assures us that the assumed uncertainties were rather conservative (values smaller than 1). Hence we do not believe that the measurement error covariance is mishandled.

P25, L14 we added: "In the current study we assumed a diagonal measurement error covariance matrix. Concerns might rise that the observational uncertainties are underestimated due to the absence of the error correlations. However the $\chi_r^2$ values prove the opposite."

**P11, L5: This equation is not referenced explicitly in the text. What does it show?**

The equation refers to the deviation term of the flux model. We clarify that by adding:

P11, L23: "the flux model (Eq.6)…."

**P11, L6 – L12: These terms are discussed before being introduced (they refer to an equation in the following subsection). I think the order needs to be changed here.**

We corrected by deleting the text. The text is moved to P14, L15.

**P11, L19: If I understand this correctly, synthetic eddy covariance (EC) data were extracted at several locations in both models, and these pseudo-fluxes were used to calculate the spatial and temporal correlation lengths for use in the inversion (please clarify that this is synthetic EC). So essentially, we are using the difference between two models as a proxy for the uncertainty correlation in the real world? I think this is fine. However, two things come to mind: 1) if we were to use "real" eddy covariance data, we would sample very much smaller length scales than the model (i.e. typically <1km, rather than 50km), so I would not expect that the derived correlations would be comparable to the same experiment using real data (as the text seems to indicate on P6); 2) since we're in model world, and in light of point (1), why not use every grid cell to calibrate the correlations? Would this come out as being very different?**

We clarify that the eddy covariance data is synthetic. We added:

P12, L15 :"Fluxes from GBIOME-BGCv1 can also be regarded as synthetic EC fluxes."

We note that the current paper describes the synthetic experiment. The aim is to perform the real data inversion (see companion paper Kountouris et al., 2016) using data driven uncertainties and for that we need a methodology. The methodology is the same for both papers based on Kountouris et al. (2015). By no means, we do want to imply that we use model-model differences as a proxy for the uncertainty in the real inversions. To evaluate correctly the system we need to apply though the same methodology but to different data sets. For the synthetic case the correct error structure will be derived by estimating the error correlations between the models which took part in the inversion. In the real world though, we perform a model-data analysis since this would be the appropriate for the real error structure. Indeed as the reviewer expected, we calculated spatial correlations significantly smaller than those derived from the model-model residual analysis. The text in P6 explicitly refers to that finding (L7). We disagree with the reviewer that P6 indicates the opposite.

Regarding the second concern the reviewer makes a good point. We have tested also the spatial correlations by calculating all the potential pixel pairs. No significant difference was found. But even if a difference was present, we selected to extract modeled fluxes only at the same locations where an EC station exist, for the sake of comparability to real data inversions.

In the companion paper we perform the same flux residual analysis for real EC data. The spatially resolved flux distribution is known only at the EC measurement sites. By selecting the same grid-cells in the synthetic case we make sure that we do not add additional information in the error structure, information that we do not have in the real world. With this approach we make sure that the synthetic case is not over constrained and hence, it is a fair experiment comparable with a real data inversion.

P12 L24 we added: "Following this approach apart from the similarity, we also ensure that results from the synthetic experiment, would be informative for a real data inversion, by using exactly the same information to characterize the prior uncertainties."

**P12, L12: The two experiments that are carried out focus on "tuning" the covariance matrix in two ways, so as to match the overall difference between the two models: B1, scale the covariance matrix uniformly; S1 add a bias. What is the reasoning for choosing only these two methods? Couldn't this mismatch be closed in several other ways, e.g. by increasing the correlation lengths or adding a "nugget" term to the diagonal elements, etc.?**

Methods like increasing the correlation lengths or adding a nugget term to the diagonal elements are already used by Lauvaux et al. (2012) and cited P6 L21. In Kountouris et al. (2015) the analysis leaves no room to assume much larger correlation scales. Indeed we could have chosen larger correlations to increase the aggregated uncertainty but at expense of the validity of the error structure.

**P14, L4-L8: Please provide a reference for these choices of data filtering.**

P16, L5 We added : "…Geels et al. (2007)". The citation is also added in P29 L27.

**P15, L13: I don't think Thompson et al., 2011 is the most appropriate reference here.**

P19, L23 we corrected: "Tarantola 2005". The Thompson citation was deleted and instead we used the following citation: "Tarantola, A.: Inverse problem theory and methods for model parameter estimation, ISBN: 0-89871-572-5, siam, 2005."

**P16, L17 – L22: The improved correlation and "variance" is simply a product of the cost function descent. This should be clarified.**

Certainly the improved statistics are directly related to the minimization of the cost function, as the Bayesian inversion balances between data constraint and prior constraint.

We have added the following in P18, L25 to remind the reader of this: "As expected from the optimization (i.e. minimization of the cost function), the…"

**P16, L23: Does "chi-squared" show us anything here that we can extend to the real world, given that the model is perfect (see general point 1 above)?**

The chi-squared values did not intend here to show anything regarding a real data inversion. This is just a metric to evaluate how well the algorithm fits the observational (dry mole fractions) data, and also to evaluate our prior error assumptions. This is a rather important step before proceeding with a real data inversion. The fact that there are no transport uncertainties does not make the whole model perfect. We refer here to the answers to the first comment.

**P17, L7: Again, isn't this a trivial result showing that the gradient descent is working?**

As explained in the very first comment, the gradient algorithm minimizes the model-data mismatch (concentration space), and not the fluxes themselves. We refer here to the answers to the first comment.

**P18, L11: Probably should be noted that this will largely be determined by the model-measurement mismatch uncertainty covariance, rather than the prior uncertainty.**

The uncertainty reduction in flux space is defined as [1 – (posterior flux uncertainty)/(prior flux uncertainty)]. The posterior uncertainty depends on the prior, the measurement and the transport uncertainty. However it is not obvious that the uncertainty reduction is largely determined by the measurement error covariance. Hence we would like to avoid this statement. Instead we clarify in the paper the dependence of the posterior uncertainties. We added:

P21, L26: "…and B1 respectively. Note that whilst the prior uncertainty refers only to the
flux space, the posterior uncertainty depends on the uncertainty of prior fluxes,
measurements, and atmospheric transport".

**P19, L15: I think this is a very strong conclusion to draw here. I'd contend that the**
**suitability of EC data for "validation" of inverse model fluxes is dominated by scaling**
**issues. In this paper, it is assumed that the EC data is representative of 50km^2. In**
**reality, EC data will sample scales that are orders of magnitude smaller.**

We agree with the reviewer that EC data is representative for much smaller scales on the
order of ~1 km$^2$. We note that the retrieved fluxes are at ~25 km resolution not 50 km.
Nevertheless, still the scales are not directly comparable. However this method is also used
in Broquet et al. (2013) where posterior flux estimates were compared against EC data.
Despite the scale mismatch they found that posterior mismatches are in good agreement
with the theoretical uncertainties.

We corrected in text the word clearly and now read:

P23, L1: "…potentially shows…"

**P20, L1: I think this shows that your inversion algorithm is working, not that you**
**would get any closer to the truth in the real world.**

We believe this is answered in the very first comment made by the reviewer.

**P21, L13: See general point 1.**

We refer to our response for the general point 1 and 2.

**P22, L11: I don't think we can comment on the reliability of the results of a real world**
**inversion here. A real world inversion will likely be dominated by chemical transport**
**model errors, which are not quantified here**.

With respect to the error assumptions and whether we underestimated the uncertainties in
the measurement error covariance, we refer to the specific comment P4, L31 from referee
3. We note that we follow a standard approach for the characterization of the transport
model uncertainties and more information can be found in Rödenbeck (2005). Inversions
are typically adding the transport error in the measurement error covariance matrix as a
diagonal element. Off diagonal elements usually are not considered, since there is no direct
method to fully characterize (spatial and temporal autocorrelation lengths) the transport error. Potentially, one could have a rough estimation of the transport uncertainty, by running a number of different transport models, and comparing the simulated atmospheric mole fractions with observations or by calculating the range of the posterior flux estimates. Since all metrics show a very good fit to the atmospheric but also to the flux data, we have no reason to reject those results. Regarding the real data inversions, if transport uncertainties are dominating, we would expect also a bad data fitting. In that case we would agree that probably the posterior flux information would not be informative at finer scales, where complex atmospheric transport patterns can not be fully captured by the atmospheric transport models (e.g. mesoscale circulations etc). However, at coarser aggregated scales fluxes would still be informative as long as the data show a good fitting performance.

P23 L17 we added: "In the current study we do not excessively assess the transport error but it is rather included as diagonal elements in the measurement error covariance, which is typical in atmospheric inversions. The chi square values confirm that there is no underestimation of the uncertainties. We note though that erroneous flux estimates are likely to be estimated, especially at finer spatial scales where the transport model is not able to resolve the real transport (e.g. individual eddys, complicate terrain etc). However, for coarser spatial scales transport models perform better, and as long as the fitting performance shows good results, flux estimates should be more reliable."

P26 L7 we added: "…at least for aggregated scales up to the country level"

---

## Author Response (AR2)

[revised manuscript text omitted]

**Response to comments raised by reviewer #3**

We thank the reviewer for his helpful comments, and hope to have addressed the concerns in the revised manuscript. Below we respond to each of the comments, and list the corresponding modifications that were made to the manuscript. The original comments are in bold face letters, and our replies are in regular face letters.

**1. My previous comment still stands that improvements in "goodness of fit" tests (RMSE, correlation, chi-squared) are simply a consequence of the inversion code performing in a way that is not obviously erroneous. In their response the authors state that an improvement in flux space is not a necessarily condition of an improvement in concentration space. Whilst this is true in the real world (in contrast to some of the comments in the revised paper, see below), it should absolutely not be the case in the situation where the model and data are perfect, as they are here. Think of the limiting case: if model-data error was very small, and prior flux error was large, one should be able to derive very close to "perfect" fluxes in the case of a perfect model and perfect data. By reducing the data constraint, one simply moves the flux solution back from "perfection" towards the prior.**

This comment refers to the previous first comment, where the reviewer was not convinced that different inversion setups can be compared using a synthetic data experiment. We agree that the metrics of RMSE and correlation mostly indicate that the inversion is working. We also agree that a comparison between different inversion setups might not be possible using a pseudo-data experiment. However, it is not our intention to actually compare different setups in order to decide which setup performs better than another. We intended to assess the spatial and temporal scales at which we can expect the inversion system to inform us about surface-atmosphere exchange fluxes, which is different from quantitatively comparing different setups. We do think that this information on the relevant spatiotemporal scales, at which the inversion is informative, can be transferred to the real-world case, at least to a certain degree.

Regarding perfect model and data we think there seems to be a fundamental misunderstanding of how the synthetic experiment is performed. Unlike the reviewer states, the model used in the inversion is not perfect (i.e. identical to the model used for the truth). The flux fields as a priori are generated from a completely different biosphere model than that used to generate the "true" fluxes. Also uncertainties for model-data mismatch are included in a realistic way. The only aspect of the synthetic experiment where the model is perfect is the atmospheric transport (i.e. the identical transport is used for forward and inverse, and no realization of the model-data mismatch error was added to the simulated concentrations), but the inversion does not make use of this "perfect" transport model, as a realistic model-data mismatch error is used. Even for the limiting case mentioned by the reviewer, where model-data mismatch error is very small and prior uncertainties are large, we would not expect the retrieved fluxes to perfectly match the "true" fluxes, as the underlying spatiotemporal patterns are fairly different between "true" and prior fluxes; this is related to the fact that the information provided by the observations is not sufficient to fully constrain the fluxes at all spatiotemporal scales (i.e. the inverse problem is underdetermined).

We attempted to clarify this in the revised manuscript in the following way:

In response also to the comment below (P15 L20), we now state in the paper:

P15 L25 "We note that the synthetic data were derived without adding realizations of model-data mismatch error"

P16 L3 we added: "The use of two different biosphere models ensures that the posterior simulated $CO_2$ time series will never exactly match the pseudo-observations (the known truth)."

Regarding the potential impact of transport model error:

P23 L24 now reads: "Note that the current study does not include the impact of transport model imperfection, as we used the identical transport model in the inversion as for generating the synthetic observations. However the inversion includes the related uncertainty in a realistic way as diagonal elements in the measurement error covariance, which is typically done in atmospheric inversions. The chi square values confirm that there is no underestimation of the uncertainties. We note though that erroneous flux estimates are likely to be estimated especially at finer spatial scales where the transport model is not able to resolve the real transport (e.g. individual eddys, complex terrain etc). However, at those coarser spatial scales, where the inversion shows good performance, transport models are also expected to perform better, thus limiting the impact of transport model errors."

**2. My second general comment highlighted several papers that had developed methodology in this area in recent years. Whilst the authors have now cite a couple of them, I was hoping that, since this is a "methodology" paper, they would compare and contrast the methodology in this paper with the recent developments by other authors. However, instead they simply say that eddy covariance data are not available for the gases used in the other publications. This misses the point to me; if these papers applied their methodology to CO2 (and there is no reason why they couldn't), would the limitations identified by the authors still apply? If so, what would be the benefits of the proposed system.**

We thank the reviewer for pointing out the importance of this comparison. We provide a more detailed information, which we hope will make this point clearer.

The abovementioned papers from Ganesan et al. (2014) and Lunt et al. (2016), are using a
hierarchical Bayesian method to infer terrestrial fluxes from dry mole tracer fractions. They
follow a method which has been used over the last 10 years and introduced first by Michalak et
al. (2004; 2005). In this approach, the so called "hyperparameters" (spatial and temporal
correlation scales which are characterizing the covariance matrix as well as mean and standard
deviation which they describe the prior Probability Density Function), are derived from
atmospheric data. The main difference between those studies is that the latter calculates the
covariance parameters first and then, in a second step, derives the fluxes. In Ganesan et al.
(2014) and Lunt et al. (2016) studies, covariance parameters and fluxes are calculated
simultaneously. Since the main principle is the same, atmospheric data are used to infer the
covariance parameters, those parameters are strongly dependent on the atmospheric data as well
as the transport model (Michalak et al. 2005). Assuming that two studies are sharing the same
atmospheric data set but different transport model, they will potentially derive a different set of
optimized parameters. Since we refer to hyperparameters within the prior error covariance (flux
uncertainties), they should not depend on the transport model.

It would be of great interest if we could directly compare the two approaches, the one described
in Ganesan et al. (2014) and the one described in the current study. Apart from the fact that this
would be beyond the scope of this study, a direct comparison is not possible since the two
approaches have been applied to different tracers. Nevertheless, the study from Michalak et al.
(2004), as explained, uses the same principle as Ganesan et al. (2014) and the companion papers.
They report correlation length scales for the prior land fluxes of around 1000 km, which is
significantly different from the lengths we derived (around 40 km, see Kountouris et al., 2015)
and which we apply in the companion paper (Kountouris et al., 2016). Whilst the prior error
covariance parameters are driven from the atmosphere in the above mentioned papers, in our
study those parameters are directly driven from the fluxes themselves.

Lunt et al. (2016) reduced the spatial dimension of the state vector, where the data are used to
determine this reduction. They achieved this by defining basis functions partitioning the domain
into Voronoi cells. However, computational cost seems to be significant and they choose to
perform their optimization only in the spatial space, ignoring the time dimension. In our study
spatial and temporal space (three-hourly fluxes) are being optimized.

P6 L10 now reads: "In a similar approach Ganesan et al. (2014) and Lunt et al. (2016), applied a
hierarchical Bayesian model using atmospheric concentrations, to estimate both, fluxes and a set
of hyper-parameters (e.g. mean and standard deviation of a priori emissions PDF as well as
model – measurement standard deviation and autocorrelation scales). Those studies are focused
on sulfur hexafluoride ($SF_6$) and methane ($CH_4$), so a direct comparison of spatial autocorrelation
scales for prior flux uncertainties in $CO_2$ inversions is not possible. In addition, Lunt et al. (2016)
reports that due to the computational costs, they performed the inversion with no temporal
dependence, assuming that the fluxes are constant over a fixed time period. Furthermore, the
covariance parameters depend on the atmospheric data and on the transport model used (Michalak et., 2005). Michalak et al. (2004) applied a similar approach to $CO_2$ and reported
spatial decorrelation lengths of around 1000 km, which are one order of magnitude larger than
our estimates. ''

**Additional comments:**

**- P15 L20 In the revised paper the authors now state that no error was added to the**
**model/data. Wouldn't it have been more interesting to include some uncertainty here and**
**see how it influenced the mismatch between the "true" and a posteriori flux. This follows**
**directly from comment 1 above. Without any error in the model/data, the improvement in**
**fit is fairly trivial.**

We think that this pointed issue is just a misunderstanding. We clarify that in atmospheric
inversions most of the synthetic studies, use the same model to derive prior fluxes and to
construct the synthetic observations. This would obviously result to a zero model data mismatch.
Hence an error realization should be added, to at least, give some mismatch between the model
and the "measurements". For example, in Lunt et al. (2016), they perform also a pseudo
experiment. However, the pseudo observations were derived simply by scaling the prior flux
distribution. Then they added a random noise to the observations to simulate model-
measurement errors.

By using a different model for the prior, and another model to construct the observations, we
make sure that the inversions have to overcome a difference in model structure. Such an
approach is already used in Tolk et al. (2011). This causes the simulated CO2 time series (prior)
to never perfectly match with the true.

P16 L1 we added: "The use of two different biosphere models assures that the prior simulated
$CO_2$ time series will never exactly match the pseudo-observations (the known truth)."

**- P23 L17, states that model transport error is not "excessively assessed". This is not true.**
**Model transport error is ignored.**

In atmospheric inversions, the measurement error covariance matrix (i.e $Q_c$ in Eq.3) accounts
typically for measurement errors, the representation error (uncertainties introduced from the
insufficient grid resolution to resolve fluxes as small scales) and the transport error. Hence
transport uncertainties are implicitly assumed in the error covariance matrix which is a standard
approach in atmospheric inversions.

P23 L21 we clarify: "Note that the current study does not include the impact of transport model
imperfection, as we used the identical transport model in the inversion as for generating the synthetic observations. However the inversion includes the related uncertainty in a realistic way as diagonal elements in the measurement error covariance, which is typically done in atmospheric inversions. The chi square values confirm that there is no underestimation of the uncertainties."

**- P23 L22 now states that "for coarser spatial scales transport morels perform better, and as long as the fitting performance shows good results, flux estimates should be more reliable." The first part of this sentence is an unproven assertion (that better resolution = better transport), and the second part is not correct (that better fit to the data = more reliable flux). I could have a terrible transport model that fits the data perfectly simply by compensating for transport error by deriving completely wrong fluxes.**

We agree with the reviewer that higher resolution models do not necessarily simulate better the atmosphere. In fact, we say the same thing: transport models, perform better at coarser spatial scales. We do not refer to higher resolution models and their potentials here. For the second part of the sentence we agree again with the reviewer that a good fit does not assure a reliable flux. However, validating the fluxes with independent measurements could be an indicator if the flux correction was done appropriately.

P23 L28 we clarify: "However, at those coarser spatial scales, where the inversion shows good performance, transport models are also expected to perform better, thus limiting the impact of transport model errors."

**- P25 L15 states that "Concerns might rise that the observational uncertainties are underestimated due to the absence of the error correlations. However the $\chi r2$ values prove the opposite." This may be true in this synthetic case, but it won't hold at all in the real world. In reality there must be correlated errors in chemical transport models.**

This statement is misleading. We already mentioned that we make use of a data density function (see section 2.1 and Rödenbeck, C., 2005). The function inflates the uncertainty and it is equivalent with the assumption that transport errors are correlated in the order of a week.

In the inversion system, data and model uncertainties result to a total uncertainty of

$$\sigma_{tot} = \sqrt{\sigma_{mod}^2 + \sigma_{obs}^2}$$

The data density inflates the uncertainty over weekly intervals and the total uncertainty is now given as

$\sigma_i = \sqrt{N^*} \cdot \sigma_{tot}$ where $N^*$ is the number of measurements within a given time interval. This way, we do consider that transport errors are correlated in time.

P10 L4 we added: "The data density inflates the uncertainty over weekly intervals by a factor of the square root of the number of measurements within a given time interval. This ensures…"

P25 L28 we clarify: "However we do consider implicitly that transport errors might by correlated over time, and we do consider that via the data density function. Further, for the synthetic study the $\chi_r^2$ values prove a fair treatment of the observational uncertainties."

**- My previous comment that the comparison with pseudo-EC data still stands, and has not been addressed in the revised manuscript. The authors cannot conclude from this study that EC data can be used to validate these fluxes in the real world. In the real world, scaling issues will be absolutely critical (EC data being representative of scales that are typically an order of magnitude smaller than the transport model).**

Although we addressed reviewer's comment in our previous response, we unintentionally did not include it in the revised manuscript. We thank the reviewer for his comment; we made the following changes to the manuscript:

P23 L3-6: we deleted: "The very close correspondence of these results with those shown in Figure 7 for the domain-wide monthly flux budget potentially shows that eddy covariance measurements can principally be used for validation of the inverse estimates at monthly timescales."

P25 L4 we added : "The "true" fluxes were used to validate the posterior flux estimates. In this synthetic experiment, both fluxes share the same spatial resolution (25km) which makes the validation straightforward. In a real data inversion, eddy covariance measurements will substitute the "true" fluxes making the spatial scales not directly comparable. Despite the scale mismatch, Broquet et al. (2013) compared the posterior flux estimates against eddy covariance data with promising results; showing that posterior mismatches are in good agreement with the theoretical uncertainties."

---

## Author Response (AR3)

[revised manuscript text omitted]

**Authors response**

We would like to sincerely thank the co-editor, for carefully reading our manuscript and for his helpful comments during the discussion phase. For sake of clarity and to easily track changes in the manuscript, the current revised version contains already the changes from the previous revised manuscripts (as accepted). We only keep track of changes here, those that refer to the editor's comments.

We revised the manuscript to properly refer to it as technical note paper. We made the following changes:

P1 L1 Title now reads: "Technical Note: Atmospheric $CO_2$ inversions at the mesoscale using data driven prior uncertainties. Methodology and system evaluation"

P8 L5 We corrected: "…in a follow up study"

P8 L7 we corrected : "In the follow up study.."

P16 L19 we corrected : "In the follow up study.."

P24 L15 we corrected: "This technical note describes…"

P24 L27 we corrected: "which is described in a follow up study"